# PhyMPGN: Physics-encoded Message Passing Graph Network for spatiotemporal PDE systems

**Bocheng Zeng[1], Qi Wang[1], Mengtao Yan[1], Yang Liu[2], Ruizhi Chengze[3], Yi Zhang[3], Hongsheng Liu[3], Zidong Wang[3], Hao Sun[1,*]**

[1]Gaoling School of Artificial Intelligence, Renmin University of China, Beijing, China
[2]School of Engineering Science, University of Chinese Academy of Sciences, Beijing, China
[3]Huawei Technologies, Shenzhen, China

Emails: `zengbocheng@ruc.edu.cn` (B.Z.); `haosun@ruc.edu.cn` (H.S.)

## Abstract

Solving partial differential equations (PDEs) serves as a cornerstone for modeling complex dynamical systems. Recent progresses have demonstrated grand benefits of data-driven neural-based models for predicting spatiotemporal dynamics (e.g., tremendous speedup gain compared with classical numerical methods). However, most existing neural models rely on rich training data, have limited extrapolation and generalization abilities, and suffer to produce precise or reliable physical prediction under intricate conditions (e.g., irregular mesh or geometry, complex boundary conditions, diverse PDE parameters, etc.). To this end, we propose a new graph learning approach, namely, Physics-encoded Message Passing Graph Network (PhyMPGN), to model spatiotemporal PDE systems on irregular meshes given small training datasets. Specifically, we incorporate a GNN into a numerical integrator to approximate the temporal marching of spatiotemporal dynamics for a given PDE system. Considering that many physical phenomena are governed by diffusion processes, we further design a learnable Laplace block, which encodes the discrete Laplace-Beltrami operator, to aid and guide the GNN learning in a physically feasible solution space. A boundary condition padding strategy is also designed to improve the model convergence and accuracy. Extensive experiments demonstrate that PhyMPGN is capable of accurately predicting various types of spatiotemporal dynamics on coarse unstructured meshes, consistently achieves the state-of-the-art results, and outperforms other baselines with considerable gains.

## 1 Introduction

Complex dynamical systems governed by partial differential equations (PDEs) exist in a wide variety of disciplines, such as computational fluid dynamics, weather prediction, chemical reaction simulation, quantum mechanisms, etc. Solving PDEs is of great significance for understanding and further discovering the underlying physical laws.

Many traditional numerical methods have been developed to solve PDEs (Ames, 2014; Anderson, 1995), such as finite difference methods, finite element methods, finite volume methods, spectral methods, etc. However, to achieve the targeted accuracy, the computational cost of these methods for large simulations is prohibitively high, since fine meshes and small time stepping are required (typically relaxed for implicit methods). In recent years, deep learning methods have achieved great success in various domains such as image recognition (He et al., 2016), natural language processing (Vaswani et al., 2017) and information retrieval (Devlin et al., 2019). A increasing number of neural-based methods have been proposed to learn the solution of PDEs.

Physical-informed neural networks (PINNs) (Raissi et al., 2019) and its variants have shown promise performance in modeling various physical systems, e.g., fluid dynamics (Gu et al., 2024; Raissi et al.,

---

*Corresponding author

2020), rarefied-gas dynamics (De Florio et al., 2021), equation discovery (Chen et al., 2021), and engineering problems (Rezaei et al., 2022; Haghighat et al., 2021; Rao et al., 2021; Eshkofti & Hosseini, 2023). However, the family of PINN methods easily encounter scalability and generalizability issues when dealing with complex systems, due to their fully connected neural network architecture and the incorporation of soft physical constraints via loss regularizer. Neural operators (Lu et al., 2021; Li et al., 2021; 2023b; Wu et al., 2023), the variants of Transformer (Geneva & Zabaras, 2022; HAN et al., 2022; Li et al., 2023a), graph neural networks (GNN) (Battaglia et al., 2018; Seo et al., 2020; Iakovlev et al., 2021; Pfaff et al., 2021; Brandstetter et al., 2022), and pretrained diffusion models (Li et al., 2024c) overcome the above issues to some extent; however, they require substantial training datasets to learn the mapping between infinite function spaces and temporal evolution, due to their "black-box" data-driven learning manner. The physics-encoded learning, e.g., PeRCNN (Rao et al., 2022; 2023), has the ability to learn spatiotemporal dynamics based on limited low-resolution and noisy measurement data. However, this method is inapplicable to irregular domains. Therefore, achieving accurate predictions of spatiotemporal dynamics based on limited low-resolution training data within irregular computational domains remains a challenge.

To this end, we propose a graph learning approach, namely, Physics-encoded Message Passing Graph Network (PhyMPGN), to model spatiotemporal PDE systems on irregular meshes given small training datasets. Specifically, our **contributions** can be summarized as follows:

- We develop a physics-encoded graph learning model with the message-passing mechanism (Gilmer et al., 2017) to model spatiotemporal dynamics on coarse (low-resolution) unstructured meshes, where the temporal marching is realized via a second-order numerical integrator (e.g. Runge-Kutta scheme).
- Considering the universality of diffusion processes in physical phenomena, we design a learnable Laplace block, which encodes the discrete Laplace-Beltrami operator, to aid and guide the GNN learning in a physically feasible solution space.
- The boundary conditions (BCs) are taken as the *a priori* physics knowledge. We propose a novel padding strategy to encode different types of BCs into the learning model to improve the solution accuracy.
- Extensive experiments show that PhyMPGN is capable of accurately predicting various types of spatiotemporal dynamics on coarse unstructured meshes with complex BCs and outperforms other baselines with considerable margin, e.g., exceeding 50% gains.

## 2 RELATED WORK

Traditional numerical methods solve PDEs via spatiotemporal discretization and approximation. However, these methods require fine meshes, small time stepping, and fully known PDEs with predefined initial conditions (ICs) and BCs. Recently, given the advance of deep learning, numerous data-driven neural-based models have been introduced to learn PDE systems with speedup inference.

**Physics-informed learning:** PINNs (Raissi et al., 2019) pioneer the foundation for the paradigm of physics-informed learning, where automatic differentiation are used to determine the PDE residual as soft regularizer in the loss function. Further, several variants (Yu et al., 2022; Eshkofti & Hosseini, 2023; Miao & Li, 2023) have been proposed to improve their accuracy and enhance the capability to handle complex geometries. In addition to automatic differentiation, other methods can be employed to compute the PDE residual, such as finite difference (Ren et al., 2022; Rao et al., 2023), finite element (Rezaei et al., 2024; Sahli Costabal et al., 2024), and finite volume methods (Li et al., 2024a;b). The family of PINN methods (Raissi, 2018; Seo & Liu, 2019; Yang & Foster, 2022; Ren et al., 2023; 2024) perform well in solving various forward and inverse problems given limited training data or even no labeled data. However, the explicit PDEs needs to be supplied.

**Neural Operators learning:** Neural operators learn a mapping between two function spaces from finite input-output data pairs, such as the ICs, BCs, and PDE parameters. Finite-dimensional operator methods (Bhatnagar et al., 2019; KHOO et al., 2020; Zhu & Zabaras, 2018; Iakovlev et al., 2021) are mesh-dependent and cannot obtain solutions at unseen nodes in the geometry. DeepONet (Lu et al., 2021), as a general neural operator, has demonstrated with generalizability to low-dimensional systems. FNO (Li et al., 2021) and its variants (George et al., 2024; Tran et al., 2023; Guibas et al., 2022; Li et al., 2023b) learn an integral kernel directly in the Fourier domain. Moreover, U-NO (Rahman et al., 2023) and U-FNO (Wen et al., 2022) incorporate UNet (Ronneberger et al., 2015) and

FNO for deeper architecture and data efficiency. LSM (Wu et al., 2023) designs a neural spectral block to learn multiple basis operators in the latent space. Generally, neural operator methods do not require any knowledge about the underlying PDEs, but demand a large amount of training data.

**Autoregressive learning:** Autoregressive learning methods model spatiotemporal dynamics iteratively. Generally, such methods employ a neural network to extract spatial patterns and a recurrent block to model temporal evolution, such as RNN (Hochreiter & Schmidhuber, 1997; Cho et al., 2014), Transformers (Vaswani et al., 2017), or a numerical integrator. PA-DGN (Seo et al., 2020) combines the spatial difference layer with a recurrent GNN to learn the underlying dynamics. Phy-CRNet (Ren et al., 2022) employs ConvsLSTM (Shi et al., 2015) to extract spatial features and evolve over time. LED (Vlachas et al., 2022) deploys RNNs with gating mechanisms to approximate the evolution of the coarse-scale dynamics and utilizes auto-encoders to transfer the information across fine and coarse scales. The Transformer architecture combined with different embedding techniques (Geneva & Zabaras, 2022; HAN et al., 2022) for physical systems has also been explored. Besides, there are also lots of neural models (Rao et al., 2023; Brandstetter et al., 2022; Pfaff et al., 2021; Choi et al., 2023; Mi et al., 2024) employing numerical integrators for temporal marching.

**Graph diffusion processes:** Diffusion processes on graphs (Freidlin & Wentzell, 1993; Freidlin & Sheu, 2000) have a various applications, such as image processing (Lozes et al., 2014; Gilboa & Osher, 2009) and data analysis (Coifman et al., 2005; Belkin & Niyogi, 2003). Recently, studies exploring the connection between these processes and GNNs have been increasing.Implementing diffusion operations on graphs enhances the representational capacity of graph learning (Atwood & Towsley, 2016; Liao et al., 2019; Gasteiger et al., 2019). The notion of the diffusion PDE on graphs also inspires the understanding and design of GNNs (Chamberlain et al., 2021; Thorpe et al., 2022).

## 3 METHODS

### 3.1 PROBLEM SETUP

Let us consider complex physical systems, governed by spatiotemporal PDEs in the general form:

$$\dot{\boldsymbol{u}}(\boldsymbol{x}, t) = \boldsymbol{F}(t, \boldsymbol{x}, \boldsymbol{u}, \nabla \boldsymbol{u}, \Delta \boldsymbol{u}, \dots) \tag{1}$$

where $\boldsymbol{u}(\boldsymbol{x}, t) \in \mathbb{R}^m$ is the vector of state variable with $m$ components we are interested in, such as velocity, temperature or pressure, defined over the spatiotemporal domain $\{\boldsymbol{x}, t\} \in \Omega \times [0, \mathcal{T}]$. Here, $\dot{\boldsymbol{u}}$ denotes the derivative with respect to time and $\boldsymbol{F}$ is a nonlinear operator that depends on the current state $\boldsymbol{u}$ and its spatial derivatives.

We focus on a spatial domain $\Omega$ with non-uniformly and sparsely observed nodes $\{\boldsymbol{x}_0, \dots, \boldsymbol{x}_{N-1}\}$ (e.g., on an unstructured mesh), presenting a more challenging scenario compared with structured grids. Observations $\{\boldsymbol{U}(t_0), \dots, \boldsymbol{U}(t_{T-1})\}$ are collected at time points $t_0, \dots, t_{T-1}$, where $\boldsymbol{U}(t_i) = \{\boldsymbol{u}(\boldsymbol{x}_0, t_i), \dots, \boldsymbol{u}(\boldsymbol{x}_{N-1}, t_i)\}$ denote the physical quantities. Considering that many physical phenomena involve diffusion processes, we assume the diffusion term in the PDE is known as *a priori* knowledge. Our goal is to develop a graph learning model with small training datasets capable of accurately predicting various spatiotemporal dynamics on coarse unstructured meshes, handling different types of BCs, and producing the trajectory of dynamics for an arbitrarily given IC.

Before further discussion, we provide some notations. Let a graph $G = (V, E)$, with node $i \in V$ denoting the observed node in the domain and undirected edge $(i, j) \in E$ denoting the connection between two nodes. We apply Delaunay triangulation to the discrete nodes to construct the non-uniform mesh, which forms the edges of the graph.

### 3.2 MODEL ARCHITECTURE

According to the method of lines (MOL) to discretize the spatial derivatives in $\boldsymbol{F}$ at these discrete nodes, Eq. 1 can be rewritten as a system of ordinary differential equations (ODEs) by numerical discretization (Schiesser, 2012; Iakovlev et al., 2021). The ODE at each node can be described by $\boldsymbol{u}(t) = \boldsymbol{u}(0) + \int_0^t \dot{\boldsymbol{u}}(\tau) d\tau$. Numerous ODE solvers can be applied to solve it, e.g., the second-order Runge-Kutta (RK2) scheme:

$$\boldsymbol{u}^{k+1} = \boldsymbol{u}^k + \frac{1}{2}(\boldsymbol{g}_1 + \boldsymbol{g}_2) \cdot \delta t; \quad \boldsymbol{g}_1 = \boldsymbol{F}(t^k, \boldsymbol{x}, \boldsymbol{u}^k, \dots); \quad \boldsymbol{g}_2 = \boldsymbol{F}(t^{k+1}, \boldsymbol{x}, \boldsymbol{u}^k + \delta t \boldsymbol{g}_1, \dots) \tag{2}$$

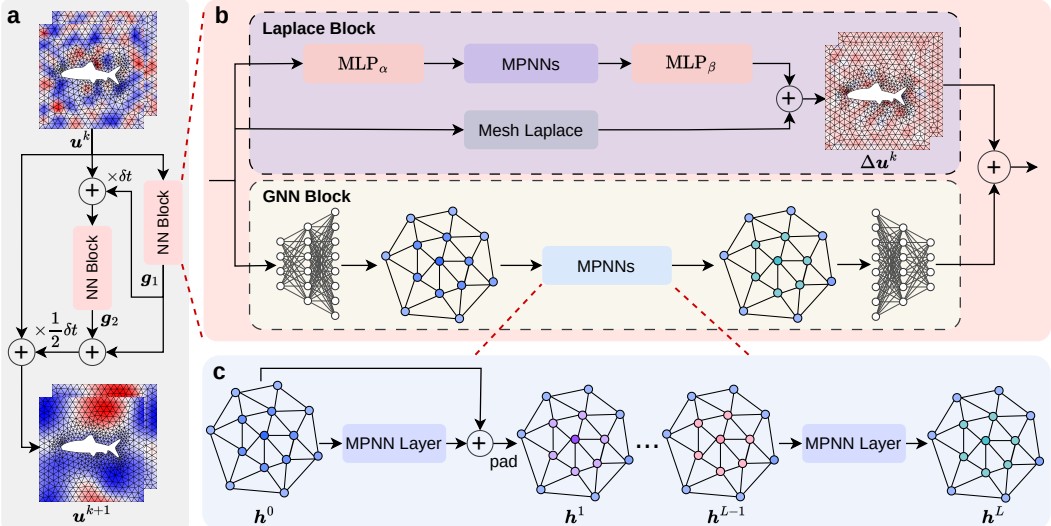

Figure 1: (a) Model with the second-order Runge-Kutta scheme. (b) NN block consists of two parts: a GNN block followed the Encode-Process-Decode framework and a Laplace block with the correction architecture. Mesh Laplace in Laplace block denotes the discrete Laplace-Beltrami operator in geometric mathematics. The other three modules, $\text{MLP}_\alpha, \text{MLP}_\beta$, and MPNNs, constitutes the lightweight learnable network for correction. (c) The MPNNs module in GNN block consists of $L$ identical MPNN layers. Residual connection and padding in latent space are applied in the first $L-1$ layers, excluding the last layer.

where $\boldsymbol{u}^k$ is the state variable at time $t^k$, and $\delta t$ denotes the time interval between $t^k$ and $t^{k+1}$.

According to the recursive scheme in Eq. 2, we develop a GNN to learn the nonlinear operator $\boldsymbol{F}$. Once the ICs are given, all subsequent states can be quickly obtained. In addition to the RK2 scheme, other numerical methods such as the Euler forward scheme and the fourth-order Runge-Kutta scheme (RK4) can also be applied, offering a trade-off between computing resource and accuracy (more details found in Appendix A). Notably, all experimental results presented in this paper are obtained using the RK2 scheme.

Figure 1a shows the architecture of our model with the RK2 scheme. The NN block aims to learn the nonlinear function $\boldsymbol{F}$ and consists of two parts (Figure 1b): a GNN block followed the Encode-Process-Decode module (Battaglia et al., 2018) and a learnable Laplace block. Due to the universality of diffusion processes in physical phenomena, we design the learnable Laplace block, which encodes the discrete Laplace-Beltrami operator, to learn the increment caused by the diffusion term $\Delta \boldsymbol{u}$ in the PDE, while the GNN block is responsible to learn the increment induced by other unknown mechanisms or sources.

### 3.2.1 GNN BLOCK

We utilize a message-passing GNN (Gilmer et al., 2017) with the Encode-Process-Decode framework (Battaglia et al., 2018) as a pivotal component of our model, referred to as the GNN block. It comprises three modules (Figure 1b–c): encoder, processor, and decoder.

**Encoder:** Within the GNN block, there are two encoders (MLPs), one for computing node embeddings and the other for edge embeddings. For each node $i$, the node encoder maps the state variable $\boldsymbol{u}_i^k$, spatial coordinate $\boldsymbol{x}_i$ and one-hot node type $\boldsymbol{C}_i$ (normal, ghost, . . . ; details shown in Section 3.3) to a node embedding $\boldsymbol{h}_i^0 = \text{NodeEnc}([\boldsymbol{u}_i^k, \boldsymbol{x}_i, \boldsymbol{C}_i])$. For each edge $(i, j)$, the edge encoder maps the relative offset $\boldsymbol{u}_{ij}^k = \boldsymbol{u}_j^k - \boldsymbol{u}_i^k$, displacement $\boldsymbol{x}_{ij} = \boldsymbol{x}_j - \boldsymbol{x}_i$, and distance $d_{ij} = \|\boldsymbol{x}_{ij}\|_2$ to an edge embedding $\boldsymbol{e}_{ij} = \text{EdgeEnc}([\boldsymbol{u}_{ij}^k, \boldsymbol{x}_{ij}, d_{ij}])$, aiming to capture local features. The concatenated features at each node and edge are encoded to high-dimensional vectors, respectively.

**Processor:** The processor consists of $L$ message passing neural network (MPNN) layers, each with its own set of learnable parameters. Consistent with the intuition of the overall model, we

expect each MPNN layer to update node embeddings in an incremental manner. Therefore, we introduce residual connections (He et al., 2016) in the first $L - 1$ layers, excluding the last layer which produces the aggregated increment we desire. The updating procedure is given as follows

$$
\begin{aligned}
\text{edge } (i, j) \text{ message:} \quad & \boldsymbol{m}_{ij}^l = \phi \left( \boldsymbol{h}_i^l, \, \boldsymbol{h}_j^l - \boldsymbol{h}_i^l, \, \boldsymbol{e}_{ij} \right), \, 0 \le l < L \\
\text{node } i \text{ update:} \quad & \boldsymbol{h}_i^{l+1} = \gamma \left( \boldsymbol{h}_i^l, \, \bigoplus_{j \in \mathcal{N}_i} \boldsymbol{m}_{ij}^l \right) + \boldsymbol{h}_i^l, \, 0 \le l < L - 1 \\
& \boldsymbol{h}_i^L = \gamma \left( \boldsymbol{h}_i^{L-1}, \, \bigoplus_{j \in \mathcal{N}_i} \boldsymbol{m}_{ij}^{L-1} \right)
\end{aligned}
\tag{3}
$$

where $\boldsymbol{h}_i^{l+1}$ denotes the embedding of node $i$ output by the $l \in [0, L)$ layer, and $\phi$ and $\gamma$ are implemented using MLPs.

**Decoder:** After $L$ MPNN layers, the decoder, which is also an MLP like the encoder, transforms the node embedding $\boldsymbol{h}^L$ in the high-dimensional latent space to the quantity in physical dimensions.

### 3.2.2 LAPLACE BLOCK

Motivated by PeRCNN (Rao et al., 2023), which uses a physics-based finite difference convolutional layer to encode prior physics knowledge of PDEs, we design a Laplace block to encode the discrete Laplace-Beltrami operators for the diffusion term $\Delta \boldsymbol{u}$ commonly seen in PDEs. This aids and guides the GNN learning in a physically feasible solution space.

Using the finite difference method to compute the Laplacian works well on regular grids, but it becomes ineffective on unstructured meshes. Therefore, we employ the discrete Laplace-Beltrami operators (Reuter et al., 2009) to compute the discrete geometric Laplacian on a non-uniform mesh domain (Pinkall & Polthier, 1996; Meyer et al., 2003), which are usually defined as

$$
\Delta f_i = \frac{1}{d_i} \sum_{j \in \mathcal{N}_i} w_{ij}(f_i - f_j) \tag{4}
$$

where $\mathcal{N}_i$ denotes the neighbors of node $i$ and $f_i$ denotes the value of a continuous function $f$ at node $i$. The weights read $w_{ij} = [\cot(\alpha_{ij}) + \cot(\beta_{ij})]/2$ (Pinkall & Polthier, 1996), where $\alpha_{ij}$ and $\beta_{ij}$ are the two angles opposite to the edge $(i, j)$. The mass is defined as $d_i = a_V(i)$ (Meyer et al., 2003), where $a_V(i)$ denotes the area of the polygon formed by connecting the circumcenters of the triangles around node $i$ (see Figure 2; more details found in Appendix B).

Figure 2: (a) Four discrete nodes ($i$, $j$, $p$, and $q$) and five edges connecting them. The two angles opposite to the edge $(i, j)$ are $\alpha_{ij}$ and $\beta_{ij}$. (b) The blue region denotes the area of the polygon formed by connecting the circumcenters (e.g., $c_1$, $c_2$, etc.) of the triangles around node $i$.

Note that Eq. 4, referred to as the Mesh Laplace module in Figure 1b, exhibits good accuracy on dense meshes but yields unsatisfactory results on coarse meshes. Therefore, we employ a lightweight neural network to rectify the Laplacian estimation on coarse meshes, which also leverages the Encode-Process-Decode framework. Consequently, Eq. 4 is updated as follows:

$$
\Delta f_i = \frac{1}{d_i} \left( z_i + \sum_{j \in \mathcal{N}_i} w_{ij}(f_i - f_j) \right) \tag{5}
$$

where $z_i$ is the output of the lightweight network, as shown in Figure 1b, where $\text{MLP}_\alpha$ and $\text{MLP}_\beta$ are employed as the encoder and the decoder, respectively, and MPNNs serve as the processor consisting of $L'$ message passing layers. The MPNNs in the Laplace block differ slightly from those in the GNN block (in particular, residual connections are applied to every layer, not just the first $L' - 1$ layers). This correction architecture of the Laplace block greatly improves the accuracy of Laplacian predictions on coarse unstructured meshes.

In summary, we approximate the nonlinear operator $\boldsymbol{F}$ by

$$
\begin{aligned}
\boldsymbol{F}(t^k, \boldsymbol{x}, \boldsymbol{u}^k, \nabla \boldsymbol{u}^k, \Delta \boldsymbol{u}^k, \dots) &\approx \text{NN\_block}(\boldsymbol{x}, \boldsymbol{u}^k) \\
&= \text{GNN\_block}(\boldsymbol{x}, \boldsymbol{u}^k) + \text{Laplace\_block}(\boldsymbol{x}, \boldsymbol{u}^k)
\end{aligned}
\tag{6}
$$

### 3.3 ENCODING BOUNDARY CONDITIONS

Given the governing PDE, the solution depends on both ICs and BCs. Inspired by the padding methods for BCs on regular grids in PeRCNN (Rao et al., 2023), we propose a novel padding strategy to encode four types of BCs on irregular domains. Specifically, we perform BC padding in both the physical and latent space. PENN (Horie & MITSUME, 2022) also constructs their model to satisfy Dirichlet and Neumman BCs by designing special neural layers and modules while we apply the padding strategy directly to the features. More details of the comparison between these methods are presented in Appendix C.

Before constructing the unstructured mesh by Delaunay triangulation, we first apply the padding strategy to the discrete nodes in the physical space (e.g., $\boldsymbol{u}^k$), as shown in Figure 3. We consider four type of BCs: Dirichlet, Neumann, Robin, and Periodic. Nodes on the Dirichlet boundary will

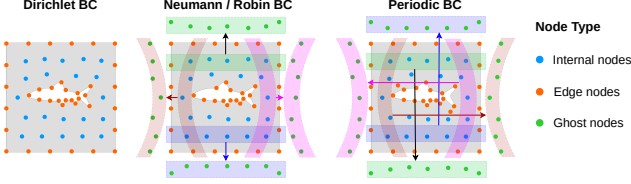

Figure 3: Diagram of boundary condition (BC) padding.

be directly assigned specific values. For Neumann/Robin BCs, ghost nodes are created symmetrically with respect to the nodes near the boundary (along the normal direction) in the physical space, and their padded values depend on derivatives in the normal direction. The goal is to ensure the true nodes, the boundary, and the ghost nodes satisfy the BCs in a manner similar to the central difference method. For periodic BCs, the nodes near the boundary are flipped and placed near the other corresponding boundary, achieving a cyclic effect in message passing. Once the padding is completed, Delaunay triangulation is applied to construct the mesh, which serves as the graph input for the model. Apart from this, we also apply padding to the prediction (e.g., $\boldsymbol{u}^{k+1}$) of the model at each time step to ensure that it satisfies BCs before being fed into the model for next-step prediction. As for the latent space (e.g., $\boldsymbol{h}^k$ in the GNN block), padding is also applied after each MPNN layer except the last layer. Details of encoding different types of BCs are provided in Appendix C.

Encoding BCs into the model indeed leads to a well-posed optimization problem. By incorporating these BCs into the learning process, the model is guided to produce solutions that adhere to the physical constraints, resulting in more accurate and physically meaningful predictions.

### 3.4 NETWORK TRAINING

The pushforward trick and the use of training noise as demonstrated in previous approaches (Brandstetter et al., 2022; Pfaff et al., 2021; Stachenfeld et al., 2022; Sanchez-Gonzalez et al., 2020) have effectively improved the stability and robustness of the model. We leverage these insights by synergistically integrating them to train our network for long-term predictions.

Due to GPU memory limitations, directly feeding the entire long time series ($T$ steps, $\boldsymbol{u}^0, \ldots, \boldsymbol{u}^{T-1}$) into the model and performing backpropagation across all steps is impractical. To address this, we segment the time series into multiple shorter time sequences of $M$ steps ($\boldsymbol{u}^{s_0}, \ldots, \boldsymbol{u}^{s_{M-1}}, T \gg M$). With the input $\boldsymbol{u}^{s_0}$, the model rolls out for $M-1$ times to generate predictions ($\hat{\boldsymbol{u}}^{s_1}, \ldots, \hat{\boldsymbol{u}}^{s_{M-1}}$), but backpropagation is only applied to the first and last predictions. Furthermore, we introduce a small noise to the first frame $\boldsymbol{u}^{s_0}$ of each segment during training. Both techniques aim to alleviate overfitting and error accumulation. Therefore, the loss function of each time segment is defined as

$$\mathcal{L} = \text{MSE}(\boldsymbol{u}^{s_1}, \hat{\boldsymbol{u}}^{s_1}) + \text{MSE}\left(\boldsymbol{u}^{s_{M-1}}, \hat{\boldsymbol{u}}^{s_{M-1}}\right) \tag{7}$$

where $\boldsymbol{u}$ denotes the ground truth of state variable, $\hat{\boldsymbol{u}}$ denotes the prediction from our model, and the superscript of $\boldsymbol{u}, \hat{\boldsymbol{u}}$ denotes their time steps in the segment.

**Evaluation metrics:** We choose the mean square error (MSE), relative $L_2$ norm error (RNE), and Pearson correlation coefficient between the prediction $\hat{\boldsymbol{u}}$ and ground truth $\boldsymbol{u}$ as the evaluation metrics, where RNE is defined as $||\hat{\boldsymbol{u}} - \boldsymbol{u}||_2 / ||\boldsymbol{u}||_2$ and correlation is defined as $\text{cov}(\boldsymbol{u}, \hat{\boldsymbol{u}}) / \sigma_{\boldsymbol{u}} \sigma_{\hat{\boldsymbol{u}}}$.

## 4 EXPERIMENTS

Here, we present comprehensive evaluation of our methods. Firstly, we validate the feasibility and efficacy of the Laplace block to compute the Laplacian on coarse non-uniform meshes. Subse-

Table 1: The MSE and RNE of four methods to approximate Laplacian on coarse domain.

| Model | Parameters | MSE | RNE |
|---|---|---|---|
| Laplace block | 5.6k | **151** | **0.033** |
| Mesh Laplace | 0 | 2574 | 0.138 |
| SDL | 21.5k | 1210 | 0.095 |
| SDL-padding | 21.5k | 408 | 0.055 |

Table 2: Datasets description.

| | Burgers | FN | GS |
|---|---|---|---|
| train/test sets | 10/10 | 3/10 | 6/10 |
| time steps | 1600 | 2000 | 2000 |
| $\delta t$ | 0.001 | 0.004 | 0.5 |
| nodes | 503 | 983 | 4225 |
| BC | periodic | periodic | periodic |

quently, the entire model is trained and evaluated on various PDE datasets and compared with several baseline models. Finally, we perform ablation studies to demonstrate the positive contribution of each individual component within our model and the impact of numerical integrator. The source code is publicly available in the Pytorch[1] and Mindspore[2] repositories.

## 4.1 LAPLACE BLOCK VALIDATION

We create a synthetic dataset to validate the effectiveness of the Laplace block. First, we generate 10,000 samples of random continuous functions on a high-resolution ($129 \times 129$) regular grid by cumulatively adding sine and cosine functions, satisfying periodic BCs. By employing the 5-point Laplacian operator of the finite difference method, the Laplacian of the functions at each node is computed as the ground truth. Then the dataset is down-sampled from the high-resolution regular grids to a set of coarse non-uniform mesh points yielding only 983 nodes. Note that 70% of the dataset is used as the training set, 20% as the validation set, and 10% as the test set. More details of the synthetic dataset are shown in Appendix D.1.

We compare the performances of four methods: (1) Laplace block with BC padding; (2) the discrete Laplace-Beltrami operator with BC padding, referred to as Mesh Laplace; (3) the spatial difference layer for Laplacian, referred to as SDL (Seo et al., 2020); (4) SDL with BC padding, referred to as SDL-padding. The MSE and RNE of all experiments are shown in Table 1. Mesh Laplace performs poorly compared to other learnable methods, while our Laplace block with the fewest parameters greatly outperforms other methods. SDL-padding exhibits an obvious performance improvement, indicating the effectiveness and importance of our padding strategy. A snapshot of a random function, ground truth of its Laplacian, and prediction are shown in Appendix Figure A.4.

## 4.2 MODELING PDE SYSTEMS

Solving PDEs serves as a cornerstone for modeling complex dynamical systems. However, in fields such as climate forecasting, reactions of new chemical matters, and social networks, the underlying governing PDEs are either completely unknown or only partially known. Hence, there is a great need for data-driven modeling and simulation. Based on the assumption, we train and evaluate our model on various physical systems, including Burgers' equation, FitzHugh-Nagumo (FN) equation, Gray-Scott (GS) equation, and flows past a cylinder, particularly in small training data regimes. We also compare our model against several representative baseline models, including DeepONet (Lu et al., 2021), PA-DGN (Seo et al., 2020), MGN (Pfaff et al., 2021), and MP-PDE (Brandstetter et al., 2022). The numbers of learnable parameters and training iterations for all models are kept as consistent as possible to ensure a fair comparison. All the data are generated using COMSOL with fine meshes and small time stepping, which are further spatiotemporally down-sampled to establish the training and testing datasets. More details of the datasets are discussed in the Appendix D.2.

**Generalization test over ICs:** We firstly test the generalizability of the model over different random ICs. For the Burgers' equation, we generate 20 trajectories of simulation data given different ICs (10/10 for training/testing). For the FN and GS equations, the number of trajectories used for training are 3 and 6, respectively. More descriptions about the datasets of these three equations are provided in Table 2). We consider periodic BCs in these PDEs. Figure 4 depicts the prediction result of each training model for the testing ICs, including the error distribution across all time steps, correlation curves and predicted snapshots at the last time step. The generalization prediction by

---

[1]https://github.com/intell-sci-comput/PhyMPGN
[2]https://gitee.com/mindspore/mindscience/tree/master/MindFlow/applications/data_mechanism_fusion/phympgn

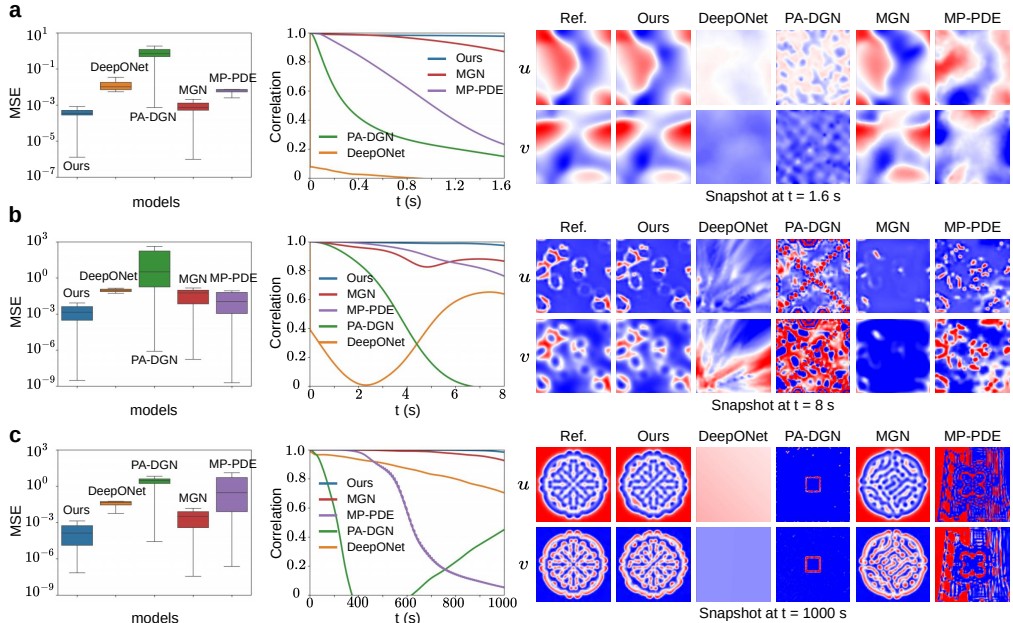

Figure 4: The error distribution (left) over time steps, correlation curves (medium), and predicted snapshots (right) at the last time step. **(a)** Burgers' equation. **(b)** FN equation. **(c)** GS equation. All systems of these PDEs have periodic BCs.

our model agrees well with the ground truth for all the three datasets. In the contrast, the baseline models yield predictions with obvious deviations from the ground truth, although MGN performs slightly better and produces reasonable prediction. To explore the dataset size scaling behavior of our model, using Burgers' equation as an example, we train our model with different numbers of trajectories: 5, 10, 20, and 50. The numerical results are presented in Table A.6, showing that as the dataset size increases, the test error decreases.

**Handling different BCs:** To validate our model's ability to encode different types of BCs, we use Burgers' equation as an example and consider three additional scenarios with irregular domain and hybrid BCs, resulting in total four scenarios, as illustrated in Figure 5: (1) **Scenario A**, a square domain with periodic BCs, as mentioned in the previous paragraph; (2) **Scenario B**, a square domain with a circular cutout, where periodic and Dirichlet BCs are applied; (3) **Scenario C**, a square domain with a dolphin-shaped cutout, where periodic and Dirichlet BCs are applied; and (4) **Scenario D**, a square domain with semicircular cutouts on both the left and right sides, where periodic and Neumann BCs are applied. Similarly, 10/10 training and testing trajectories are generated and down-sampled with nearly $N = 600$ observed nodes in each domain. Figure 6 depicts the error distribution over all time steps, correlation curves, and predicted snapshots at the last time step, for the last three scenarios (results of Scenario A are shown in Figure 4a). It is seen that our model significantly outperforms baseline models and shows the efficacy of handling PDE systems with different BCs.

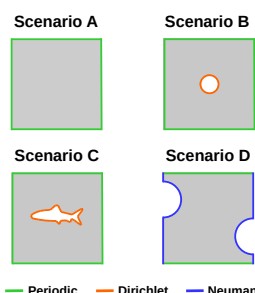

Figure 5: Four scenarios with different domains and hybrid BCs.

**Generalization test over the Reynolds number:** We also evaluate the ability of our model to generalize over the inlet velocities for the cylinder flow, governed by the Naiver-Stokes (NS) equations. Here, the Reynolds number is defined as $Re = \rho U_m D/\mu$, where $\rho$ is the fluid density, $D$ the cylinder diameter, and $\mu$ the fluid viscosity. With these parameters fixed, generalizing over the inlet velocities $U_m$ also means to generalize across different $Re$ values. By changing the inlet velocity $U_m$, we generate 4 trajectories with $Re = [160, 240, 320, 400]$ for training and 9 trajectories for testing, with only about $N = 1,600$ observed nodes in the domain. The testing sets are divided into three groups based on $Re$: 3 trajectories with small $Re = [200, 280, 360]$, 3 trajectories with medium $Re = [440, 480, 520]$, and 3 trajectories with large $Re = [600, 800, 1000]$. During the eval-

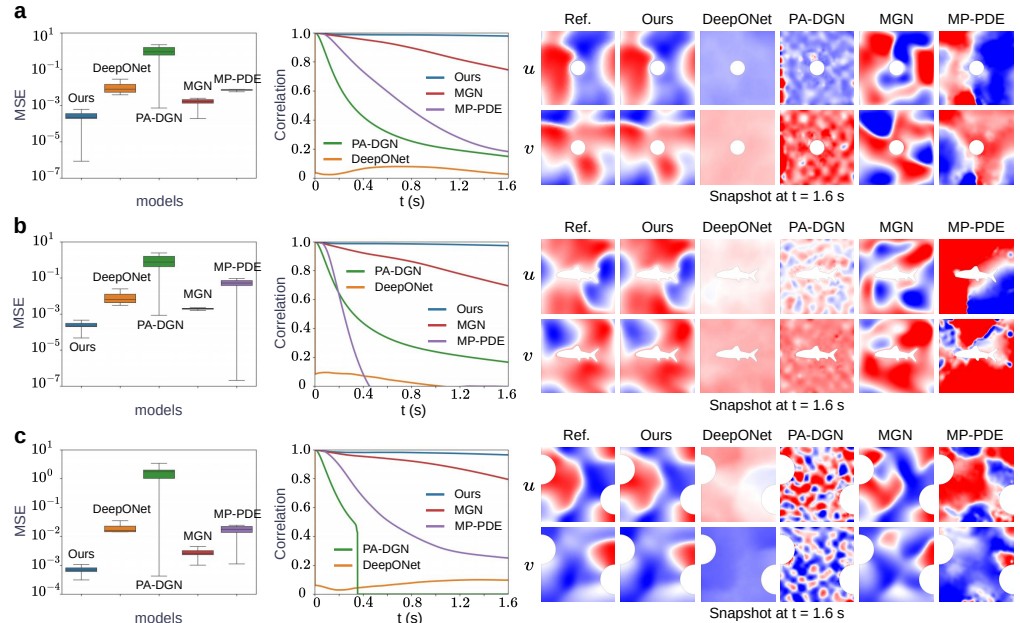

Figure 6: The error distribution (left) over time steps, correlation curves (medium), and predicted snapshots (right) at the last time step, for the Burgers' equation. **(a)** Periodic and Dirichlet BCs. **(b)** Periodic and Dirichlet BCs. **(c)** Periodic and Neumann BCs.

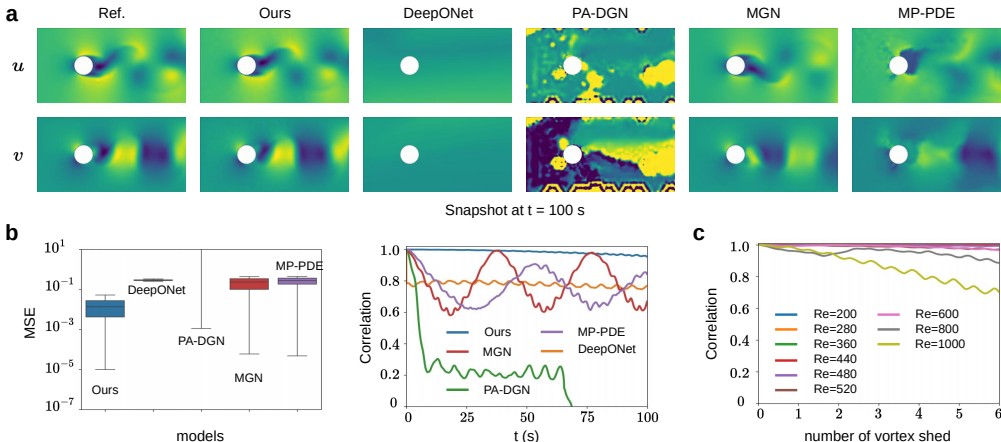

Figure 7: **(a)**–**(b)** Predicted snapshots at the last time step, error distribution over all time steps, and correlation curves for the cylinder flow problem ($Re = 480$). **(c)** Correlation curves of our model on generalization over $Re$. The number of vortex sheds within $t$ seconds is defined as $t/T_{Re}$, where $T_{Re}$ is the vortex shedding period.

uation, it is observed that the baselines struggle to generalize over $Re$ values unseen during training (e.g., the larger the $Re$, the greater the prediction error). For example, the predicted snapshots at $t = 100$ s along with the error distributions and correlation curves for $Re = 480$ are shown in Figure 7a–b. Our model consistently maintains a high correlation between the prediction and ground truth over time, while the performance of the baselines rapidly declines. It is interesting that MGN's and MP-PDE's correlation initially declines but then recovers, suggesting that they capture the cyclic dynamics but fail to generalize to bigger $Re$ values, leading to mis-prediction of the vortex shedding period. Besides, we evaluate our model's generalizability over all other $Re$ values. Considering that the larger the $Re$, the shorter vortex shedding period $T_{Re}$ and the more vortex sheds $t/T_{Re}$ in the same time span $t$, we track the prediction performance against the number of vortex sheds. Figure 7c shows the generalization result of our model across different $Re$ values, where the correlation decays faster only for $Re = 1000$. A denser set of observed nodes could help mitigate this issue.

To evaluate the extrapolation capability, our model trained over the time range $0 - 100$s is tested on the 200-second trajectory with $Re = 480$. We observe a slow decline of our model's correlation in Appendix Figure A.5. The correlation is 0.95 at $t = 100$s and decreases to 0.85 at $t = 200$s.

**Computational efficiency:** The computational cost of the Laplace block and the padding strategy is minimal. The computation of Mesh Laplace in the Laplace block includes only matrix multiplication, while the cost of the padding strategy involves merely copying the corresponding nodes in the domain and is small. A detailed analysis and comparison with MGN and MP-PDE as well as the classical numerical method (COMSOL) are presented in Appendix E. In all experiments, our model outperforms the baselines with considerable margins, e.g., exceeding 50% gains (see the specific numerical results reported in Appendix Tables A.5, A.7 and A.8). The predicted trajectory snapshots of our model and ground truth are provided in Appendix Figure A.6.

### 4.3 ABLATION STUDY

We study the effectiveness and contribution of two key components in our model, namely, the learnable Laplace block and the BC padding strategy. Specifically, we conducted ablation studies on the cylinder flow problem, comparing the performance

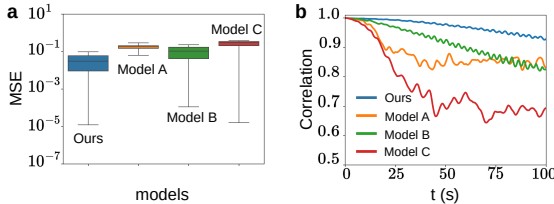

Figure 8: Results of Models A, B, C, and Ours.

of four models: (1) the full model, referred to as Ours; (2) the model only without the Laplace block, referred to as Model A; (3) the model only without the padding strategy, referred to as Model B; (4) the model without both the Laplace block and the padding strategy, referred to as Model C. Notably, the trainable parameters of the Laplace block comprise only about 1% of the entire model, and the padding strategy itself introduces no additional trainable parameters. In other words, the number of trainable parameters in these four models is almost the same. We train these models on the same cylinder flow dataset described previously and evaluate them on the testing datasets with $Re = [440, 480, 520]$. Figure 8 shows the error distribution and correlation curves, averaged over the testing datasets. The corresponding numerical MSEs of these four models are listed in Table A.9. It is seen that the ablated models exhibit a deteriorated performance (Model C performs the worst). Despite their minimal contribution to the overall trainable parameter count, the two key components greatly enhance the model's capacity for long-term predictions.

Additionally, we investigate the impact of three types of numerical integrators employed in our model on performance: Euler forward, RK2, and RK4 scheme. The MSEs of these models are presented in Table A.11, showing that the higher the order of the integrator, the better the model's prediction accuracy. However, higher-order integrators also come with greater memory and computational costs. Moreover, empirical evidence suggests that RK4 may increase the difficulty of training, including issues such as training instability. Therefore, we choose RK2 to strike a balance between computational resources and accuracy. Notably, even with Euler scheme, PhyMPGN's accuracy remains superior to baseline models with a big margin. We also explore the impact of the time segment size $M$ on results and computational requirements. Details are presented in Appendix E.

## 5 CONCLUSION

We present a graph learning approach, namely, PhyMPGN, for predicting spatiotemporal PDE systems on coarse unstructured meshes given small training datasets. Specifically, we develop a physics-encoded message-passing GNN model, where the temporal marching is realized via a second-order numerical integrator (e.g. Runge-Kutta scheme). The *a priori* physics knowledge is embedded to aid and guide the GNN learning in a physically feasible solution space with improved solution accuracy, via introducing (1) a learnable Laplace block that encodes the discrete Laplace-Beltrami operator, and (2) a novel padding strategy to encode different types of BCs. Extensive experiments demonstrate that PhyMPGN outperforms other baseline models with considerable margins, e.g., exceeding 50% gains across diverse spatiotemporal dynamics, given small and sparse training datasets. However, several challenges remain to be addressed: (1) how to effectively encode Neumann/Robin BCs in latent space; (2) the approach cannot be applied to other nonlinear or high-order diffusion terms (e.g., $\nabla^4$); (3) extending the model to 3-dimensional scenarios. These challenges highlight key directions for our future research.

ACKNOWLEDGMENTS

The work is supported by the National Natural Science Foundation of China (No. 62276269 and No. 92270118), the Beijing Natural Science Foundation (No. 1232009), and the Fundamental Research Funds for the Central Universities (No. 202230265).

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

# APPENDIX

## A    NUMERICAL INTEGRATOR

Let us consider complex physical systems, governed by spatiotemporal PDEs of the general form:

$$\dot{\boldsymbol{u}}(\boldsymbol{x}, t) = \boldsymbol{F}(t, \boldsymbol{x}, \boldsymbol{u}, \nabla \boldsymbol{u}, \Delta \boldsymbol{u}, \dots) \tag{A.1}$$

where $\boldsymbol{u}(\boldsymbol{x}, t) \in \mathbb{R}^m$ is the vector of state variable with $m$ components we are interested in, such as velocity, temperature or pressure, defined over the spatiotemporal domain $\{\boldsymbol{x}, t\} \in \Omega \times [0, \mathcal{T}]$. Here, $\dot{\boldsymbol{u}}$ denotes the derivative with respect to time and $\boldsymbol{F}$ is a nonlinear operator that depends on the current state $\boldsymbol{u}$ and its spatial derivatives.

According to the method of lines (MOL), Eq. A.1 can be rewritten as a system of ordinary differential equations (ODEs) by numerical discretization. And the ODE at each node can be described by

$$\boldsymbol{u}(t) = \boldsymbol{u}(0) + \int_0^t \dot{\boldsymbol{u}}(\tau) d\tau \tag{A.2}$$

Numerous ODE solvers can be applied to solve it, such as Euler forward scheme

$$\begin{aligned} \boldsymbol{u}^{k+1} &= \boldsymbol{u}^k + \boldsymbol{g}_1 \cdot \delta t \\ \boldsymbol{g}_1 &= \boldsymbol{F}(t, \boldsymbol{x}, \boldsymbol{u}^k, \dots) \end{aligned} \tag{A.3}$$

where $\boldsymbol{u}^k$ is the state variable at time $t^k$, and $\delta t$ denotes the time interval between $t^k$ and $t^{k+1}$. While Euler forward scheme is a first-order precision method, other numerical methods with higher precision such as Runge-Kutta schemes can also be applied, offering a trade-off between computing resource and accuracy. And the second-order Runge-Kutta (RK2) scheme can be described by

$$\boldsymbol{u}^{k+1} = \boldsymbol{u}^k + \frac{1}{2}\delta t(\boldsymbol{g}_1 + \boldsymbol{g}_2), \tag{A.4}$$

where

$$\begin{aligned} \boldsymbol{g}_1 &= \boldsymbol{F}(t^k, \boldsymbol{x}, \boldsymbol{u}^k, \dots), \\ \boldsymbol{g}_2 &= \boldsymbol{F}(t^{k+1}, \boldsymbol{x}, \boldsymbol{u}^k + \delta t \boldsymbol{g}_1, \dots) \end{aligned} \tag{A.5}$$

and the fourth-order Runge-Kutta (RK4) scheme is as followed

$$\boldsymbol{u}^{k+1} = \boldsymbol{u}^k + \frac{1}{6}\delta t(\boldsymbol{g}_1 + 2\boldsymbol{g}_2 + 2\boldsymbol{g}_3 + \boldsymbol{g}_4) \tag{A.6}$$

where

$$\begin{aligned} \boldsymbol{g}_1 &= \boldsymbol{F}(t^k, \boldsymbol{x}, \boldsymbol{u}^k, \dots), \\ \boldsymbol{g}_2 &= \boldsymbol{F}(t^k + \frac{1}{2}\delta t, \boldsymbol{x}, \boldsymbol{u}^k + \delta t\frac{\boldsymbol{g}_1}{2}, \dots), \\ \boldsymbol{g}_3 &= \boldsymbol{F}(t^k + \frac{1}{2}\delta t, \boldsymbol{x}, \boldsymbol{u}^k + \delta t\frac{\boldsymbol{g}_2}{2}, \dots), \\ \boldsymbol{g}_4 &= \boldsymbol{F}(t^k + \delta t, \boldsymbol{x}, \boldsymbol{u}^k + \delta t\boldsymbol{g}_3, \dots) \end{aligned} \tag{A.7}$$

## B    DISCRETE LAPLACE-BELTRAMI OPERATORS

Using the finite difference method to compute the Laplacian works well on regular grids, but it becomes ineffective on unstructured meshes. Therefore, we employ the discrete Laplace-Beltrami operators to compute the discrete geometric Laplacian on non-uniform mesh domain, which are usually defined as

$$\Delta f_i = \frac{1}{d_i} \sum_{j \in \mathcal{N}_i} w_{ij}(f_i - f_j) \tag{A.8}$$

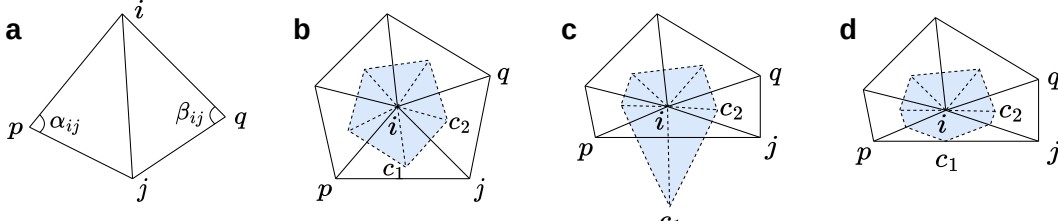

Figure A.1: (a) Illustration of four discrete nodes ($i$, $j$, $p$, and $q$) and five edges connecting them. The two angles opposite to the edge $(i, j)$ are $\alpha_{ij}$ and $\beta_{ij}$. (b) The blue region denotes the area of the polygon formed by connecting the circumcenters (e.g., $c_1$, $c_2$, and etc.) of the triangles around node $i$. (c) The scenario involves an obtuse-angled triangle $(i, p, j)$ adjacent to node $i$ with the blue region extending beyond the boundaries of these triangles. (d) The circumcenter of the obtuse-angled triangle $(i, p, j)$ is replaced by the midpoint of edge $(p, j)$, resulting in a new polygon known as the mixed Voronoi region.

where $\mathcal{N}_i$ denotes the neighbors of node $i$ and $f_i$ denotes the value of a continuous function $f$ at node $i$. The weights can be described as

$$w_{ij} = \frac{\cot(\alpha_{ij}) + \cot(\beta_{ij})}{2} \tag{A.9}$$

where $\alpha_{ij}$ and $\beta_{ij}$ are the two angles opposite to the edge $(i, j)$. The mass can be defined as $d_i = a_V(i)$, where $a_V(i)$ denotes the area of the polygon formed by connecting the circumcenters of the triangles around node $i$ (i.e., the Voronoi region, shown in Figure A.1). It is noteworthy that if there is an obtuse-angled triangle $(i, p, j)$ adjacent to node $i$, its circumcenter will extend beyond itself. In this scenario, the circumcenter of the obtuse-angled triangle $(i, p, j)$ is replaced by the midpoint of edge $(p, j)$, resulting in a new polygon known as the mixed Voronoi region.

## C  ENCODING BOUNDARY CONDITIONS

Given the governing PDE, the solution depends on both ICs and BCs. Inspired by the padding methods for BCs on regular grids in PeRCNN, we propose a novel padding strategy to encode different types of BCs on irregular domains. Specifically, we perform BC padding in both the physical space and the latent space.

Before constructing the unstructured mesh by Delaunay triangulation, we first apply the padding strategy to the discrete nodes in the physical space (i.e., $\boldsymbol{u}^k$), as shown in Figure A.2. We consider four types of BCs: Dirichlet, Neumann, Robin, and periodic. All nodes on the Dirichlet boundary will be directly assigned with specified values. For Neumann/Robin BCs, ghost nodes are created symmetrically with respect to the nodes near the boundary (along the normal direction of the boundary) in the physical space, and their padded values depends on derivatives in the normal direction. The goal is to ensure the true nodes, the boundary, and the ghost nodes satisfy the BCs in a manner similar to the central difference method. For periodic BCs, the nodes near the boundary $\Gamma_{p_1}$ are flipped and placed near the corresponding boundary $\Gamma_{p_2}$, achieving a cyclic effect in message passing (detailed formulations of various BCs are provided in Table A.2). Once the padding is completed, Delaunay triangulation is applied to construct the mesh, which serves as the graph input for the model. Apart from this, we also apply padding to the prediction (i.e., $\boldsymbol{u}^{k+1}$) of the model at each time step to ensure that it satisfies the BCs before being fed into the model for next-step prediction, as shown in Figure A.3.

As for the latent space (i.e., $\boldsymbol{h}^k$ in GNN block), padding is also applied after each MPNN layer except the last layer. For Dirichlet BCs, the embedding features of nodes on the boundary from the node encoder will be stored as the specified features for padding. For Neumann BCs with zero flux, ghost nodes are created to be symmetric with the nodes near the boundary in both physical and latent spaces. For other Neumann cases and Robin BCs, padding in the latent space remains an unsolved challenge, which is expected to be addressed in the future. And for periodic BCs, nodes near the boundary are flipped to the other side, along with their embedding features.

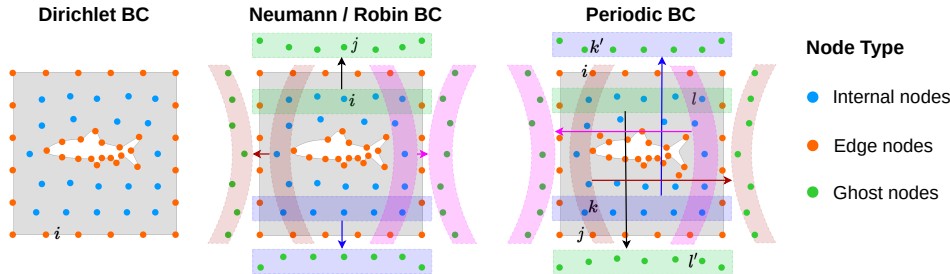

Figure A.2: Diagram of boundary condition (BC) padding

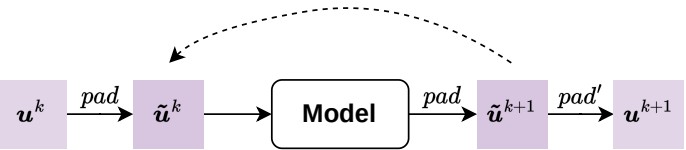

Figure A.3: Padding during the model rollout

PENN and our method both encode BCs. However, there are several key differences between the two approaches, namely,

- **Methods:** PENN designs the special neural layers and modules for the model to satisfy BCs while we develop a padding strategy directly applied to the features.

- **Types of BCs:** PENN proposes DirichletLayer and pseudoinverse decoder for Dirichlet BCs, and NeumannIsoGCN for Neumann BCs. In contrast, our padding strategy can be applied to four types of BCs (Dirichlet, Neumman, Robin, periodic), which can be seamlessly integrated into any GNN model and thus has a better generality.

- **Accuracy:** Both PENN and our padding strategy are designed to completely satisfy Dirichlet BCs without any error. However, neither method can enforce other BCs without error.

- **Efficiency:** During training, the additional cost for PENN to satisfy BCs involves constructing the pseudo-inverse decoder after the parameters are updated. The efficiency of this process depends on the number of parameters in the encoder. In contrast, the cost of our padding strategy involves mere copying the corresponding nodes in the domain during both training and testing, which is small.

To quantify the time overhead of our padding strategy, taking Burgers' trajectories with 1600 time steps and 503 nodes in the domain as an example, we compare the inference time of our models with and without the padding strategy. As shown in Table A.1, the results demonstrate that the overhead introduced by the padding strategy is small.

## D    DATASET DETAILS

### D.1    SYNTHETIC DATASET FOR LAPLACE BLOCK

We create a synthetic dataset to validate the effectiveness of the Laplace block. First, we generator 10,000 samples of random continuous functions on a high-resolution ($129 \times 129$) regular grid by cumulatively adding sine and cosine functions, satisfying periodic BCs. The formula of the random

Table A.1: The inference time of models with and without the padding strategy.

| | Inference time (s) |
|---|---|
| PhyMPGN (padding) | 10.71 |
| PhyMPGN (without padding) | 10.02 |

Figure A.4: Snapshot of a random function $f$, ground truth of its Laplacian and prediction by our Laplace block.

continuous functions is as followed

$$\tilde{f}(x,y) = \sum_{i,j=0}^{N} A_{ij}\sin\left(2\pi\left(\left(i - \left\lfloor\frac{N}{2}\right\rfloor\right)x + \left(j - \left\lfloor\frac{N}{2}\right\rfloor\right)y\right)\right) +$$
$$B_{ij}\cos\left(2\pi\left(\left(i - \left\lfloor\frac{N}{2}\right\rfloor\right)x + \left(j - \left\lfloor\frac{N}{2}\right\rfloor\right)y\right)\right) \quad\quad (A.10)$$
$$f(x,y) = \frac{\tilde{f}(x,y)}{\max \tilde{f}(x,y)}$$

where $N = 12$, $A_{ij}, B_{ij} \sim \mathcal{N}(0,1)$, and $\lfloor\cdot\rfloor$ denotes the floor function in mathematics. The function $f(x,y)$ exhibits a period of 1, satisfying periodic BCs on the computational domain $[0,1] \times [0,1]$.

By setting various random seeds, we generated a set of 10,000 samples of the random continuous function. Then, we compute the Laplacian at each node for all samples as the ground truth by employing the 5-point Laplacian operator of the finite difference method:

$$L_\Delta = \frac{1}{12(\delta x)^2}\begin{bmatrix} 0 & 0 & -1 & 0 & 0 \\ 0 & 0 & 16 & 0 & 0 \\ -1 & 16 & -60 & 16 & -1 \\ 0 & 0 & 16 & 0 & 0 \\ 0 & 0 & -1 & 0 & 0 \end{bmatrix} \quad\quad (A.11)$$

Note that 70% of the dataset is used as the training set, 20% as the validation set, and 10% as the test set. To achieve spatial downsampling, we generate a much coarser mesh by COMSOL and obtain the low-resolution simulation by choosing the closest nodes between the high-resolution and coarser mesh points. This approach is consistently applied throughout the paper for all downsampling tasks. Figure A.4 shows the snapshot of a random function, ground truth of its Laplacican and prediction by our Laplace block with $N = 983$ observed nodes.

## D.2 SIMULATION DATASET OF PDE SYSTEMS

We train and evaluate PhyMPGN's performance on various physical systems, including Burgers' equation, FitzHugn-Nagumo (FN) equation, Gray-Scott (GS) equation and cylinder flow. All the simulation data for PDE systems are generated with fine meshes and small time stepping using COMSOL, a multiphysics simulation software based on the advanced numerical methods. Subsequently, the simulation data is downsampled on both time and space before being fed into the model for training and evaluation. All the simulation data details for PDE systems are summarized in Table A.3 and Table A.4.

Table A.2: Let $\Omega$ the physical domain, $\Gamma_d, \Gamma_n, \Gamma_r$ are Dirichlet, Neumann, and Robin boundaries, respectively. And $\Gamma_{p1}, \Gamma_{p2}$ are each other's periodic boundaries. $B_d, B_n, B_r, B_{p1}, B_{p2}$ represent the set of nodes on the boundary of Dirichlet, Neumann, Robin, and Periodic, respectively. For Neumann and Robin BC, the node $i, j$ have been indicated in Figure A.2. The node $k$ is located at the intersection of the segment ij and the boundary. If such a node does not exist on the discrete point cloud, we can obtain it by interpolation or other means. For Periodic BC, nodes near the boundary $\Gamma_{p1}$ are flipped to the other boundary $\Gamma_{p2}$, achieving a cyclic effect in message passing.

| Name | Continuous Form | Discrete Form | Ghost Nodes | Padding Location (Nodes) | Padding Value |
|---|---|---|---|---|---|
| Dirichlet | $u(\boldsymbol{x}) = \bar{u}(\boldsymbol{x}), \boldsymbol{x} \in \Gamma_d$ | $u_i = \bar{u}_i, i \in B_d$ | No | Edge | $u_i = \bar{u}_i$ |
| Neumann | $\dfrac{\partial u(\boldsymbol{x})}{\partial \boldsymbol{n}} = f(\boldsymbol{x}), \boldsymbol{x} \in \Gamma_n$ | $\dfrac{u_j - u_i}{2\delta x} = f_k, k \in B_n$ | Yes | Ghost | $u_j = u_i + 2\delta x f_k$ |
| Robin | $\alpha u(\boldsymbol{x}) + \beta \dfrac{\partial u(\boldsymbol{x})}{\partial \boldsymbol{n}} = g(\boldsymbol{x}), \boldsymbol{x} \in \Gamma_r$ | $\alpha u_k + \beta \dfrac{u_j - u_i}{2\delta x} = g_k, k \in B_r$ | Yes | Ghost | $u_j = \dfrac{2\delta x}{\beta}(g_k - \alpha u_k) + u_i$ |
| Periodic | $u(\boldsymbol{x}_1) = \bar{u}(\boldsymbol{x}_2), \boldsymbol{x}_1 \in \Gamma_{p1}, \boldsymbol{x}_2 \in \Gamma_{p2}$ | $u_i = u_j, i \in B_{p1}, j \in B_{p2}$ | Yes | Ghost | $u'_k = u_k, u_p = u_l$ |

**Burgers' equation** Burgers' equation is a fundamental nonlinear PDE to model diffusive wave phenomena, whose formulation is described by

$$\dot{\boldsymbol{u}} = \nu\Delta\boldsymbol{u} - \boldsymbol{u} \cdot \nabla\boldsymbol{u} \tag{A.12}$$

where the diffusion coefficient $\nu$ is set to $5 \times 10^{-3}$.

There are four different scenarios of Burgers' equation: (1) **Scenario A**, a square domain with periodic BCs; (2) **Scenario B**, a square domain with a circular cutout, where periodic and Dirichlet BCs are applied; (3) **Scenario C**, a square domain with a dolphin-shaped cutout, where periodic and Dirichlet BCs are applied; and (4) **Scenario D**, a square domain with semicircular cutouts on both the left and right sides, where periodic and Neumann BCs are applied. The diagram of irregular domain and hybrid BCs for the four scenarios are illustrated in Figure 5. ICs for Burgers' equation are generated similarly to the synthetic dataset for the Laplace block (Appendix D.1). We generated 20 trajectories with different ICs for each scenario, using 10 for training and 10 for testing. To explore the dataset size scaling behavior of our model, we generated 50 trajectories with different ICs in total for training. Each trajectory consists of 1600 time steps with a time interval of $\delta t = 0.001$ for the model.

**FN equation** FitzHugh-Nagumo (FN) equation is an important reaction-diffusion equation, described by

$$\begin{aligned}
\dot{u} &= \mu_u\Delta u + u - u^3 - v + \alpha \\
\dot{v} &= \mu_v\Delta v + (u - v)\beta
\end{aligned} \tag{A.13}$$

where the diffusion coefficients $\mu_u$ and $\mu_v$ are respectively 1 and 10, and the reaction coefficients $\alpha$ and $\beta$ are respectively 0.01 and 0.25.

The system, governed by FN equation and defined on a square computational domain with periodic BCs, begins with random Gaussian noise for warmup. After a period of evolution, time trajectories are extracted to form the dataset. We generated 13 trajectories with different ICs, using 3 for training and 10 for testing. Each trajectory consists of 2000 time steps with a time interval of $\delta t = 0.004$ for the model.

**GS equation** Gray-Scott (GS) equation is a mathematical model used to describe changes in substance concentration in reaction-diffusion systems, which is governed by

$$\begin{aligned}
\dot{u} &= \mu_u\Delta u - uv^2 + \mathcal{F}(1 - u) \\
\dot{v} &= \mu_v\Delta v + uv^2 - (\mathcal{F} + \kappa)v
\end{aligned} \tag{A.14}$$

where the diffusion coefficients $\mu_u$ and $\mu_v$ are respectively $2.0 \times 10^{-5}$ and $5.0 \times 10^{-6}$, while the reaction coefficients $\mathcal{F}$ and $\kappa$ are respectively 0.04 and 0.06.

The system, governed by GS equation and defined on a square computational domain with periodic BCs, starts the reaction from random positions. The time trajectories of the reaction' process are extracted to form the dataset. We generated 16 trajectories with different ICs, using 6 for training and 10 for testing, Each trajectories consists of 2000 time steps with a time interval of $\delta t = 0.5$ for the model.

**Cylinder flow** The dynamical system of two-dimensional cylinder flow is governed by Navier-Stokes equation

$$\dot{\boldsymbol{u}} = -\boldsymbol{u} \cdot \nabla\boldsymbol{u} - \frac{1}{\rho}\nabla p + \frac{\mu}{\rho}\Delta\boldsymbol{u} + \boldsymbol{f} \tag{A.15}$$

where the fluid density $\rho$ is 1, the fluid viscosity $\mu$ is $5.0 \times 10^{-3}$ and the external force $\boldsymbol{f}$ is 0. We are focusing on to generalize over the inflow velocities $U_m$ of fluids while keeping the fluid density $\rho$, the cylinder diameter $D = 2$, and the fluid viscosity $\mu$ constant. According to formula $Re = \rho U_m D/\mu$, generalizing over the inlet velocities $U_m$ also means to generalize across different Reynolds numbers.

The upper and lower boundaries of the cylinder flow system are symmetric, while the cylinder surface has a no-slip boundary condition. The left boundary serves as the inlet, and the right boundary as the outlet. We generated 13 trajectories with different $Re$ values, using 4 trajectories with $Re = [160, 240, 320, 400]$ for training and 9 trajectories for testing. The testing sets are divided

Table A.3: Details of simulation data of four PDE systems.

|  | Burgers | FN | GS | Cylinder Flow |
|---|---|---|---|---|
| Domain | $[0, 1]^2$ | $[0, 128]^2$ | $[0, 1]^2$ | $[0, 16] \times [0, 8]$ |
| Original nodes | 12659 | 15772 | 37249 | 36178 |
| Down-sampled nodes | 503 | 983 | 4225 | 1598 |
| Original $\delta t$ | 0.0005 | 0.002 | 0.25 | 0.025 |
| Down-sampled $\delta t$ | 0.001 | 0.004 | 0.5 | 0.05 |
| Time steps | 1600 | 2000 | 2000 | 2000 |
| Train / Test sets | 10 / 10 | 3 / 10 | 6 / 10 | 4 / 9 |

Table A.4: Details of simulation data of four Burgers' scenarios.

|  | Scenario A | Scenario B | Scenario C | Scenario D |
|---|---|---|---|---|
| Domain | $[0, 1]^2$ | $[0, 1]^2$ | $[0, 1]^2$ | $[0, 1]^2$ |
| Original nodes | 12659 | 14030 | 14207 | 15529 |
| Down-sampled nodes | 503 | 609 | 603 | 634 |
| Original $\delta t$ | 0.0005 | 0.0005 | 0.0005 | 0.0005 |
| Down-sampled $\delta t$ | 0.001 | 0.001 | 0.001 | 0.001 |
| Time steps | 1600 | 1600 | 1600 | 1600 |
| Train / Test sets | 10 / 10 | 10 / 10 | 10 / 10 | 10 / 10 |

Table A.5: The MSEs of the three PDE systems.

|  | Burgers | FN | GS |
|---|---|---|---|
| DeepONet | 1.68e-2 | 9.47e-2 | 4.04e-2 |
| PA-DGN | 8.66e-1 | 7.19e+2 | 2.93e+0 |
| MGN | 9.19e-4 | 5.54e-2 | 4.44e-3 |
| MP-PDE | 5.88e-3 | 2.88e-2 | 2.82e+0 |
| PhyMPGN (Ours) | **2.99e-4** | **2.82e-3** | **4.05e-4** |
| Lead ↑ | 67.5% | 90.2% | 90.9% |

Table A.6: The MSEs of our model on different dataset size effect (Burgers).

| **No. of trajectories** | 5 | 10 | 20 | 50 |
|---|---|---|---|---|
| PhyMPGN (ours) | 6.35e-4 | 2.99e-4 | 1.60e-4 | 4.43e-5 |

into three groups based on $Re$: 3 trajectories with small $Re = [200, 280, 360]$, 3 trajectories with medium $Re = [440, 480, 520]$, and 3 trajectories with large $Re = [600, 800, 1000]$. Each trajectory consists of 2000 time steps with a time interval of $\delta t = 0.05$ for the model, except for the trajectories with larger Reynolds numbers ($Re = [600, 800, 1000]$), which have a time interval of $\delta t = 0.005$.

## E  MODELING PDE SYSTEMS

We train and evaluate PhyMPGN's performance on various physical systems, including Burgers' equation, FN equation, GS equation and cylinder flow. Leveraging four NVIDIA RTX 4090 GPUs, the training process is completed within a range of 4 to 15 hours. And the inference time is under one minute on a single GPU. We also compare the performance of PhyMPGN against existing approaches, such as DeepONet, PA-DGN, MGN, MP-PDE.

Table A.7: The MSEs of Burgers' equation with various boundary conditions.

|  | Scenario A | Scenario B | Scenario C | Scenario D |
|---|---|---|---|---|
| DeepONet | 1.68e-2 | 1.37e-2 | 1.16e-2 | 2.22e-2 |
| PA-DGN | 8.66e-1 | 1.10e+0 | 1.03e+0 | Nan |
| MGN | 9.19e-4 | 1.68e-3 | 1.89e-3 | 2.70e-3 |
| MP-PDE | 5.88e-3 | 6.59e-3 | 5.11e-2 | 1.63e-2 |
| PhyMPGN (Ours) | **2.99e-4** | **2.89e-4** | **3.06e-4** | **9.30e-4** |
| Lead ↑ | 67.5% | 82.8% | 83.8% | 65.6% |

Table A.8: The MSEs of cylinder flow for three groups of $Re$ values.

| $Re$ | $[200, 280, 360]$ | $[440, 480, 520]$ | $[600, 800, 1000]$ |
|---|---|---|---|
| DeepONet | 9.07e-2 | 3.10e-1 | 1.53e+0 |
| PA-DGN | 7.58e+3 | Nan | Nan |
| MGN | 1.73e-1 | 2.17e-1 | 8.88e-1 |
| MP-PDE | 7.23e-3 | 2.77e-1 | 5.96e-1 |
| PhyMPGN (Ours) | **2.14e-4** | **3.56e-2** | **1.25e-1** |
| Lead ↑ | 97.0% | 83.4% | 79.0% |

**Generalization test over ICs:** To test the models' generalizability over different random ICs, we train PhyMPGN and baseline models on the three physical systems: Burgers' equation, FN equation and GS equation. All these PDE systems have a square domain with periodic BCs. The MSEs of PhyMPGN and baseline models are listed in Table A.5. The value of "Lead" in the tables is defined as $(\text{MSE}_s - \text{MSE}_b)/\text{MSE}_s$, where $\text{MSE}_b$ denotes the best results among all models and $\text{MSE}_s$ denotes the second-best result. To explore the dataset size scaling behavior of our model, using Burgers' equation as an example, we train our model with different numbers of trajectories: 5, 10, 20, and 50. The numerical results are presented in Table A.6.

**Handling different BCs:** To validate our model's ability to encode different types of BCs, we use Burgers' equation as an example and consider three additional scenarios with irregular domain and hybrid BCs, resulting in a total of four scenarios, as illustrated in Figure 5: (1) **Scenario A**, a square domain with periodic BCs, as mentioned in the previous paragraph; (2) **Scenario B**, a square domain with a circular cutout, where periodic and Dirichlet BCs are applied; (3) **Scenario C**, a square domain with a dolphin-shaped cutout, where periodic and Dirichlet BCs are applied; and (4) **Scenario D**, a square domain with semicircular cutouts on both the left and right sides, where periodic and Neumann BCs are applied. The MSEs of PhyMPGN and baseline models are listed in Table A.7.

**Generalization test over the Reynolds number:** We also evaluate the ability of our model to generalize over the Reynolds number for the cylinder flow, governed by the Naiver-Stokes (NS) equations. We generate 4 trajectories with $Re = [160, 240, 320, 400]$ for training, and 9 trajectories for testing. The testing sets are divided into three groups based on $Re$: 3 trajectories with small $Re = [200, 280, 360]$, 3 trajectories with medium $Re = [440, 480, 520]$, and 3 trajectories with large $Re = [600, 800, 1000]$. The MSEs for the three groups of PhyMPGN and baseline models are listed in Table A.8. To evaluate the extrapolation capability, our model trained over the time range $0 - 100$s is tested on the 200-second trajectory with $Re = 480$. We observe a slow decline of our model's correlation in Appendix Figure A.5. The correlation is 0.95 at $t = 100$s and decreases to 0.85 at $t = 200$s.

**Computational efficiency:** The computational cost of the Laplace block and the padding strategy is minimal. The computation of Mesh Laplace in Laplace block includes only matrix multiplication $L_w \boldsymbol{f}$, where $\boldsymbol{f}$ are the node features and $L_w$ is a weight matrix related to mesh geometric that can

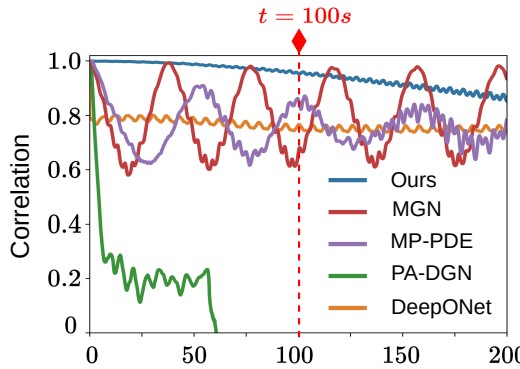

Figure A.5: The correlation of our model trained on the 100-second trajectories but tested on the 200-second trajectory with $Re = 480$.

Table A.9: The MSEs averaged on the testing sets with $Re = [440, 480, 520]$ of the four models in ablation study: Ours, Model A, B, and C.

| Model | MSE |
|---|---|
| PhyMPGN (Ours) | **3.56e-2** |
| Model A | 1.57e-1 |
| Model B | 1.06e-1 |
| Model C | 2.50e-1 |

Table A.10: The inference time (seconds) and MSE of four methods for Burgers' equation.

| | Time per trajectory | Total time (10 trajectories) | MSE |
|---|---|---|---|
| MGN | 4.01 | 5.19 | 9.19e-4 |
| MP-PDE | **0.33** | **1.00** | 5.88e-3 |
| COMSOL | 19.5 | 195 | **4.91e-5** |
| PhyMPGN | 10.71 | 13.64 | 2.99e-4 |

Table A.11: The MSEs of our models employing different numerical integrators.

| Model | $Re = [200, 280, 360]$ | $Re = [440, 480, 520]$ | $Re = [600, 800, 1000]$ |
|---|---|---|---|
| PhyMPGN (Euler) | 1.95e-3 | 1.07e-1 | 1.41e-1 |
| PhyMPGN (RK2) | 2.14e-4 | 3.56e-2 | 1.25e-1 |
| PhyMPGN (RK4) | **1.21e-4** | **1.97e-2** | **9.26e-2** |

be calculated offline. The cost of the padding strategy involves merely copying the corresponding nodes in the domain and is small.

For a comparative analysis of computational efficiency, we select MGN and MP-PDE as baselines from neural-based methods, and COMSOL as a baseline from classical numerical methods. Taking Burgers' trajectories with 1600 time steps and 503 nodes in the domain as an example, we compare the inference times between these three baselines and our model (PhyMPGN). The inference time and MSE of these four methods are shown in Table A.10. All evalutions are conducted on a single NVIDIA RTX 4090 GPU and an i9-13900 CPU. Additionally, the ground truth is generated with high resolution in both time and space using COMSOL with the implicit scheme.

Table A.12: The training and inference time of our models employing different types of numerical integrators for cylinder flow problems.

| | Training time | Inference time per trajectory | Inference time (5 trajectories) |
|---|---|---|---|
| PhyMPGN (Euler) | 5.2 h | 8.98 s | 13.47 s |
| PhyMPGN (RK2) | 14.4 h | 17.57 s | 27.08 s |
| PhyMPGN (RK4) | 39.3 h | 34.99 s | 53.49 s |

Table A.13: The MSE and the training time with different segment sizes $M$.

| $M$ | $Re = [200, 280, 360]$ | $Re = [440, 480, 520]$ | $Re = [600, 800, 1000]$ | Training time |
|---|---|---|---|---|
| 10 | 2.43e-3 | 1.48e-1 | 2.42e-1 | 8.3 h |
| 15 | 9.34e-4 | **3.05e-2** | 1.34e-1 | 15.0 h |
| 20 | **2.14e-4** | 3.56e-2 | **1.25e-1** | 14.4 h |

MGN predicts one step given one step as input, similar to PhyMPGN. However, due to the additional Laplace block, padding strategy, and RK2 scheme, PhyMPGN has more inference time than MGN. On the other hand, MP-PDE predicts $T_w$ steps (with $T_w = 20$) in a single forward pass using $T_w$ steps as input, resulting in the least inference time among neural-based models. Overall, these neural-based models can greatly improve efficiency by simultaneously inferring multiple trajectories. The results from COMSOL, a multiphysics simulation software based on advanced numerical methods, presented in the comparison, are obtained using an implicit scheme.

In summary, PhyMPGN achieves the lowest error among neural-based models with a reasonable time overhead as a trade-off, while COMSOL delivers the least error among these four methods but at the cost of significantly higher time overhead. Notably, to simulate the PDE system, COMSOL requires the complete governing PDE formulas, whereas the three neural-based methods do not, as they learn the dynamics from data.

**Ablation study:** We study the effectiveness and contribution of two key components in our model, namely, the learnable Laplace block and the BC padding strategy. Specifically, we conducted ablation studies on the cylinder flow problem, comparing the performance of four models: (1) the full model, referred to as Ours; (2) the model only without the Laplace block, referred to as Model A; (3) the model only without the padding strategy, referred to as Model B; (4) the model without both the Laplace block and the padding strategy, referred to as Model C. We train these models on the same cylinder flow dataset described previously and evaluate them on the same testing sets. The error distribution and correlation curves of these four models averaged on the testing sets with $Re = [440, 480, 520]$ are shown in Figure 8. The corresponding MSEs are listed in Table A.9.

**Investigation on numerical integrators:** Additionally, using the same datasets as before, we investigate the impact of three types of numerical integrators employed in our model on performance: the Euler forward scheme, the RK2 scheme, and the RK4 scheme. The MSEs of our models employing these three numerical integrators are presented in Table A.11. We trained our model for 1600 epochs using four NVIDIA RTX 4090 GPUs and performed inference for 2000 time steps per trajectory on a single GPU. Their training and inference time are listed in Table A.12. All inference times are under one minute. Thanks to the parallel capabilities of GPUs, the inference speed can be further accelerated by processing batches simultaneously.

**The impact of the time segment size $M$:** Many previous methods (Seo et al., 2020; HAN et al., 2022; Geneva & Zabaras, 2022; Ren et al., 2022; Rao et al., 2023) unroll models to predict multiple steps within a short time segment then backpropagate to learn the dynamics of physics systems. Intuitively, longer time segments can be advantageous for capturing long-term dependencies, thereby improving the model's long-term prediction capabilities. However, excessively long segments may introduce challenges during training, such as increased memory consumption and difficulties in

Table A.14: The number of trainable parameters of all models.

|  | Burgers | FN | GS | Cylinder Flow |
|---|---|---|---|---|
| DeepONet | 219k | 295k | 422k | 948k |
| PA-DGN | 194k | 194k | 260k | 1025k |
| MGN | 175k | 175k | 267k | 1059k |
| MP-PDE | 193k | 193k | 253k | 1069k |
| PhyMPGN (Ours) | 157k | 161k | 252k | 950k |

achieving convergence. To investigate the impact of the segment size $M$ on the results and computational requirements, we conduct experiments using the cylinder flow datasets in the ablation study. We train our model with segment sizes $M = 10, 15, 20$, and the MSEs on the testing sets and the training time are presented in Table A.13, which shows that the bigger the segment size $M$, the better the model's prediction accuracy. Additionally, since we need to adjust the batch size to fit the GPU memory as the segment size $M$ varies, the training time does not scale linearly with $M$. Specifically, the batch sizes for $M = 15$ and $M = 20$ are the same, while the batch size for $M = 10$ is double that of the other two configurations. Therefore, the training time differences are also reasonable.

In all experiments, the numbers of trainable parameters and training iterations for all models are kept as consistent as possible to ensure a fair comparison, as listed in the Table A.14. Snapshots of ground truths and predictions of $u$ component from our model in all experiments are shown in Figure A.6.

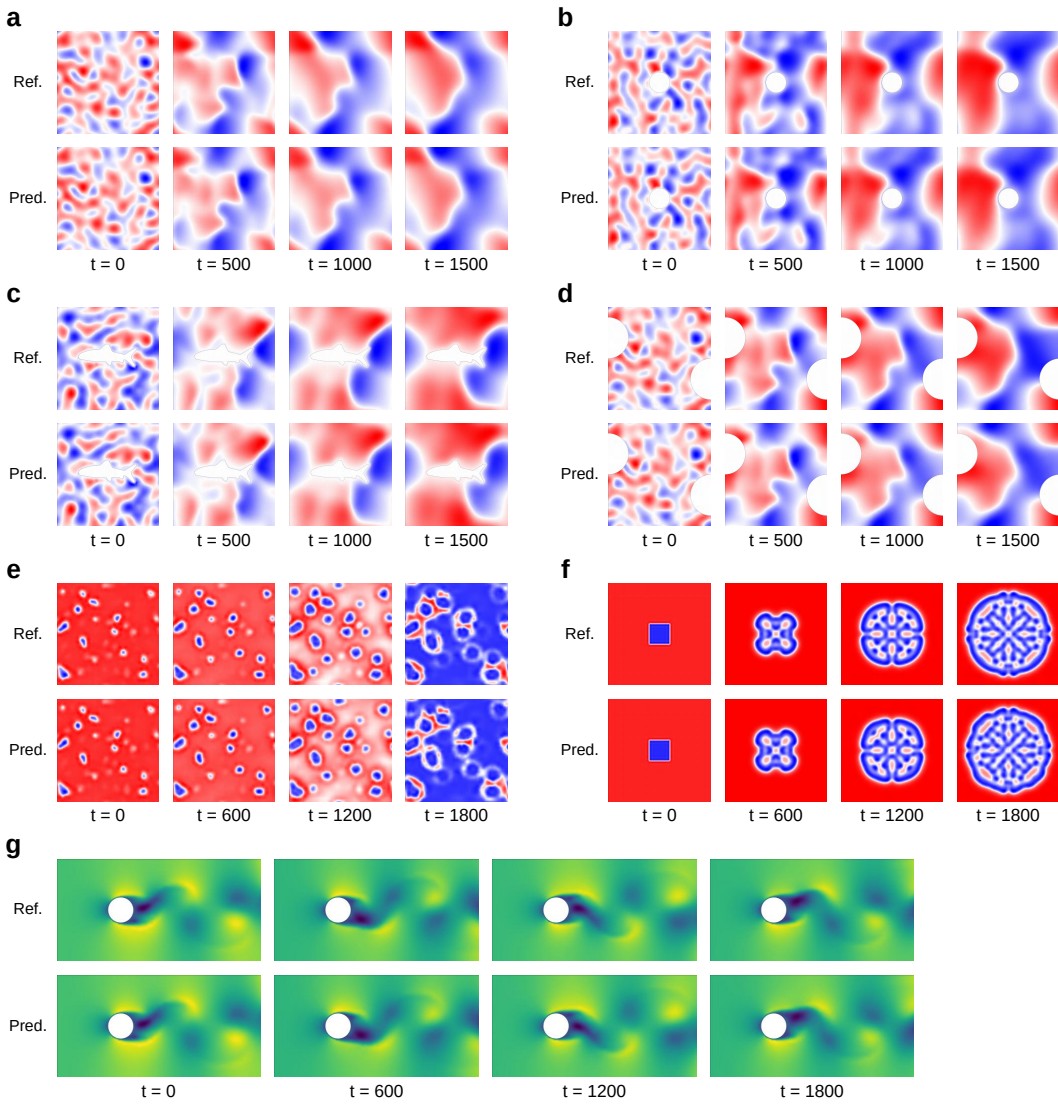

Figure A.6: Snapshots of $u$ component of ground truths in several simulations and predictions from our model. **(a, b, c, d)** Burgers' equation in various domains. **(e)** FN equation. **(f)** GS equation. **(g)** Cylinder flow with $Re = 440$.

