# OpenReview forum: "PhyMPGN: Physics-encoded Message Passing Graph Network for spatiotemporal PDE systems"
_ICLR.cc/2025/Conference — ICLR 2025 Spotlight_

### Official Review · Reviewer_zATM · 2024-10-19

**Soundness:** 3
**Presentation:** 4
**Contribution:** 3
**Rating:** 8
**Confidence:** 4

**Summary:**

The paper introduces a graph learning approach called Physics-encoded Message Passing Graph Network (PhyMPGN) designed to model spatiotemporal PDE systems on coarse unstructured meshes using small training datasets. Authors incorporate a learnable Laplace block that encodes the discrete Laplace-Beltrami operator to constrain the GNN learning within a physically feasible solution space. Additionally, a boundary condition padding strategy is proposed to handle different types of boundary conditions, enhancing model convergence and accuracy. Extensive experiments demonstrate that PhyMPGN accurately predicts various spatiotemporal dynamics on coarse unstructured meshes with complex BCs, outperforming other baseline models with considerable gains.

**Strengths:**

1.	The integration of physics-encoded components within a message-passing GNN addresses the challenges of modeling spatiotemporal dynamics on unstructured meshes.
2.	The learnable Laplace block that encodes the discrete Laplace-Beltrami operator ensures solutions remain within a physically feasible space.
3.	The experiments are thorough, showcasing the model's ability to generalize across various spatiotemporal dynamics. The ablation studies further strengthen the validity of the proposed methods.
4.	The model demonstrates strong generalization capabilities despite being trained on small datasets.

**Weaknesses:**

1.	The authors state that traditional numerical methods require fine meshes and small time stepping. This overlooks implicit schemes, which can handle larger time steps while maintaining stability.
2.	The paper does not provide a comparative analysis of the computational speed or efficiency of PhyMPGN.
3.	The rationale behind focusing solely on coarse meshes is not explained.

**Questions:**

1.	Could the authors clarify their statement regarding the limitations of traditional numerical methods? How does their approach compare with implicit numerical schemes in terms of stability and time-stepping requirements?
2.	Can the authors provide insights into the computational performance of their method?
3.	What are the main obstacles in generalizing PhyMPGN to three-dimensional problems? How might the current architecture need to adapt to handle the increased complexity?

---

> ### Author Response · Authors · 2024-11-21
> **Reply to Reviewer zATM (Part 1)**
>
> Thank you for your constructive comments and suggestions!
>
> > **Q1. Comparative analysis of the computational efficiency of PhyMPGN.**
>
> **Reply:** Good suggestion! we would like to clarify that the computational cost of the Laplace block and the padding strategy is minimal. The computation of Mesh Laplace in Laplace block includes only matrix multiplication, while the cost of the padding strategy involves merely copying the corresponding nodes in domain and is essentially negligible.
>
> For comparative analysis of computational efficiency, we select the well-known MGN [1] and MP-PDE [2] as baseline neural-based methods, and COMSOL as a baseline for classical numerical methods.
>
> MGN predicts one step given one step as input, similar to PhyMPGN. However, due to the additional Laplace block, padding strategy, and RK2 scheme, PhyMPGN has more inference time than MGN. On the other hand, MP-PDE predicts $T_w$ steps (with $T_w = 20$) in a single forward pass using $T_w$ steps as input, resulting in the least inference time among neural-based models. Overall, these neural-based models can greatly improve efficiency by simultaneously inferring multiple trajectories. The inference time and MSE of these four methods evaluated on Burgers' trajectories with 503 nodes are presented in **Table A** below.
>
> In summary, PhyMPGN achieves the lowest error among neural-based models with a reasonable time overhead as a trade-off, while COMSOL delivers the least error among these four methods but at the cost of significantly higher time overhead. Notably, to simulate the PDE system, COMSOL requires the complete governing PDE formulas, whereas the three neural-based methods do not, as they learn the dynamics from data. These comparison results have been included in Section 4.2 (page 10), Appendix E (page 24), and Appendix Table A.9 (page 25) in the revised paper.
>
> **Table A: The inference time (seconds) and MSE of four methods.**
> |   | Time per trajectory | Total time (10 trajectories) | MSE |
> | -------- | -------- | -------- | -------- |
> | MGN | 4.01 | 5.19 | 9.19e-4 |
> | MP-PDE | 0.33 | 1.00 | 5.88e-3 |
> | COMSOL | 19.5 | 195 | 4.91e-5 |
> | PhyMPGN | 10.71 | 13.64 | 2.99e-4 |
>
> > **Q2. Clarify the statement and compare with implicit numerical schemes in terms of stability and time-stepping requirements.**
>
> **Reply:** Excellent suggestion! First of all, we apologize for the inaccurate statement that traditional numerical methods require fine meshes and small time-stepping, which overlooks implicit schemes. This have been clarified in Section 1 (page 1) in the revised paper.
>
> Taking Burgers' equation with 503 mesh nodes in the domain as an example, we compare the stability and time-stepping requirements between PhyMPGN and COMSOL using an explicit scheme (RK4) and an implicit scheme (backward difference), although this is beyond the scope of our paper.
>
> Notably, PhyMPGN employs the RK2 scheme, trained with $\delta t=0.001$ previously, and is directly used to predict with larger time-steppings without retraining. The stability and MSE results of these three methods are summarized in **Table B**.
>
> We are encouraged by PhyMPGN's good stability with larger time-stepping $\delta t=0.004, 0.008$ and even $0.040$, which is comparable with COMSOL using the implicit method and remains more stable compared with COMSOL which uses the RK4 scheme. At the same time, we acknowledge that implicit schemes offer clear advantages in handling large $\delta t$. We will consider to integrate implicit methods with neural-based models in our future work.
>
> **Table B: The stability and MSE of three methods on increasing $\delta t$.**
> | dt | COMSOL (RK4) | COMSOL (implicit) | PhyMPGN |
> | -------- | -------- | -------- | -------- |
> | 0.001     | stable (4.91e-5)     | stable (4.91e-5)     | stable (2.99e-4) |
> | 0.004     | stable (5.33e-5)     | stable (5.22e-5)     | stable (2.99e-4) |
> | 0.008     | unstable (non-convergence) | stable (6.49e-5) | stable (2.98e-4) |
> | 0.040     | unstable (non-convergence)     | stable (3.72e-4) | stable (3.77e-4) |
> | 0.050     | unstable (non-convergence)     | unstable (use smaller $\delta t$ automatically)     | unstable (3.72e+0)|
>
>
> ***References:***
>
> [1] Pfaff et al., ICLR 2021.
>
> [2] Brandstetter et al., ICLR 2022.

---

> ### Author Response · Authors · 2024-11-21
> **Reply to Reviewer zATM (Part 2)**
>
> > **Q3. Rationale behind focusing solely on coarse meshes.**
>
> **Reply:** Great remark! First, modeling dynamics on coarse meshes presents a more challenging task. Coarse meshes have fewer observation nodes in the domain, which results in less information about the system's dynamics compared to fine meshes. In practical scenarios, we may not always have access to a large number of high-resolution observation in the domain.
>
> Second, simulation on coarse meshes is typically more efficient than on fine meshes, especially for large-scale computational tasks. This is particularly true for complex systems with large spatial and temporal scales, such as weather forecasting. The coarse grid method provides a practical solution for these large-scale simulations.
>
> > **Q4. Generalizing PhyMPGN to three-dimensional problem.**
>
> **Reply:** Insightful question! The Mesh Laplace module in our Laplace block is a discrete differential geometry operator designed for triangulated 2-manifolds [3]. This makes it applicable to 2D scenarios and 3D surfaces, but not to 3D volumes. For 3D PDE systems, alternative methods (such as cubic FEM) to compute the Laplacian on irregular domains need to be incorporated into the models, which sets forward our future work. Thanks for your suggestion!
>
> ***Reference:***
>
> [3] Meyer et al.,  Visualization and Mathematics III 2003.

---

> ### Author Response · Authors · 2024-11-23
> **Looking forward to your feedback**
>
> Dear Reviewer zATM,
>
> Again, thanks for your constructive comments. We would like to follow up on our rebuttal to ensure that all concerns have been adequately addressed. If there are any further questions or points that need discussion, we will be happy to address them. Your feedback is invaluable in helping us improve our work, and we eagerly await your response.
>
> Thank you very much for your time and consideration.
>
> Best regards,
>
> The Authors

---

> ### Comment · Reviewer_zATM · 2024-11-23
> **Thank you for your answers**
>
> Dear authors, thank you for addressing my comments. I don't have any additional questions.
> Update: having considered the excellent work done by the authors on the responses to the other reviewers, not only mine, I have decided to raise my score.

---

> ### Author Response · Authors · 2024-11-27
> **Thank you for your feedback**
>
> Dear Reviewer zATM,
>
> We sincerely appreciate your attention, recognition of our work, and responsible attitude. Your time and effort dedicated to reviewing our paper are deeply valued!
>
> Best regards,
>
> The Authors

---

### Official Review · Reviewer_m5AJ · 2024-10-30

**Soundness:** 4
**Presentation:** 4
**Contribution:** 4
**Rating:** 10
**Confidence:** 5

**Summary:**

I love the paper for its outperforming MGN; it has a solid baseline by DeepMind! It introduces PhyMPGN, a Physics-encoded Message Passing Graph Network that effectively models spatiotemporal PDE systems on irregular meshes using limited data. By integrating physics through a learnable Laplace-Beltrami operator and a novel boundary condition padding strategy, the approach ensures physically accurate predictions. PhyMPGN significantly outperforms existing methods, achieving over 50% performance gains and demonstrating strong generalization across various PDEs and conditions. The model's efficiency and robustness make it a valuable advancement for scientific simulations where data is sparse or complex geometries are involved.

**Strengths:**

Outperform a solid baseline; writing is good, presentation is good. The method is clear and new. The strength of this paper lies in its innovative integration of physics-based knowledge into graph neural networks, which enables accurate and efficient modeling of complex spatiotemporal PDE systems on irregular meshes with limited data. By employing a learnable Laplace block and a novel boundary condition padding strategy, PhyMPGN ensures solutions remain physically consistent and precise, overcoming limitations of traditional and purely data-driven methods. Additionally, the model's demonstrated ability to generalize well to different PDEs, geometries, and conditions, along with its significant performance gains over existing techniques, highlights its robustness and versatility for real-world scientific applications

**Weaknesses:**

There is some need for some clarification when it is for convection-dominant problem.

**Questions:**

I have several questions to help me understand the method thoroughly. Firstly, how will the model behave if we delete (or do not use) the Mesh Laplace? Will it still be better than MGN? Secondly, why second-order RK for MOL? What about RK4 or forward Euler? Do these different choices matter? Thirdly, I understand graph Laplace preserves some diffusion physics, which is excellent and universal enough. Still, for examples of inviscid first order PDE such as wave/ advection equation, or inviscid Burger equation,  Euler equations, will add graph Laplacian make the traveling feature blur?

---

> ### Author Response · Authors · 2024-11-21
> **Reply to Reviewer m5AJ**
>
> Thank you for the positive feedback and constructive comments. We are also grateful that you recognized the strengths and contributions of our work.
>
> > **Q1. Model's behavior when deleting the Mesh Laplace.**
>
> **Reply:** Excellent question! Similar to the ablation study in Section 4.3 (page 10), the model without the Mesh Laplace module, referred to as Model D, is trained on the same cylinder flow datasets and evaluated on the testing sets with $Re = [440, 480, 520]$. The MSEs of Model D and the other four ablation models are listed in **Table A** below. It can be seen that the MSE of Model D is nearly identical to that of Model A. This is because the number of trainable parameters in Model A and Model D is nearly the same.
>
> The ablated models include:
>
> - the full model, referred to as **Ours**.
> - the model only without the Laplace block, referred to as **Model A**.
> - the model only without the padding strategy, referred to as **Model B**.
> - the model without both the Laplace block and the padding strategy, referred to as **Model C**.
> - the model only without the Mesh Laplace, referred to as **Model D**.
>
> Model D's MSE $1.62\times10^{-1}$ is still lower than MGN's $2.17\times 10^{-1}$, which can be attributed to our padding strategy. MGN entirely relies on neural network to learn patterns that satisfy BCs, which inevitably leads to errors accumulating during the roll-out process, particularly when training datasets are small. In contrast, our proposed padding strategy enforces strict adherence with BCs, helping constrain the solution space and accurately simulate physical systems for long term.
>
> **Table A: The MSEs averaged on the testing sets with $Re=[440, 480, 520]$ of the five models for ablation study: Ours, Model A, B, C and D.**
> | Model | MSE |
> | -------- | -------- |
> | PhyMPGN (Ours)    | 3.56e-2     |
> | Model A     | 1.57e-1     |
> | Model B     | 1.06e-1     |
> | Model C     | 2.50e-1     |
> | Model D     | 1.62e-1     |
>
> > **Q2. Integrator choices between Euler, RK2, and RK4.**
>
> **Reply:** Good question! Euler forward scheme consumes less memory and computational cost but has lower accuracy, whereas RK4 consumes larger memory and computational cost but offers higher accuracy. Additionally, empirical evidence suggests that RK4 may increase the difficulty of training, including issues such as training instability. Therefore, we choose RK2 to strike a balance between computational resources and accuracy.
>
> As shown in Section 4.3 (line 517-523 in the revised paper), we investigate the impact of three types of numerical integrators employed in our model on performance: Euler, RK2, and RK4. The MSEs of our models employing these three numerical integrators are presented in Appendix Table A.9 (page 25), showing that the higher the accuracy of the integrator, the better the model’s prediction accuracy.
>
> > **Q3. Convection-dominant problems and inviscid PDE systems.**
>
> **Reply:** Insightful question! As mentioned in Section 3.2 (page 3), the Laplace block in our model is specifically designed for the diffusion term in PDE systems, while the GNN block is tasked with learning the dynamics governed other unknown mechanisms or sources. For convection-dominant problems, the GNN block plays an more dominant role than the Laplace block. For inviscid systems, the Laplace block should not be encoded into the model, and the GNN block has to take full responsibility for learning the dynamics. However, the boundary conditions (BCs) can still be effectively encoded using our padding strategy, which helps constrain the solution space and accurately simulate the systems for long term.
>
> In summary, we appreciate your constructive comments and suggestions. Please let us know if you have any other questions. We look forward to your feedback!

---

> > ### Comment · Reviewer_m5AJ · 2024-11-21
> > **Thank you.**
> >
> > The authors addressed my comments.

---

> > > ### Author Response · Authors · 2024-11-22
> > > **Thank you for your feedback**
> > >
> > > Dear Reviewer m5AJ,
> > >
> > > Thank you very much for your positive feedback. Your time and effort placed on reviewing our paper are highly appreciated!
> > >
> > > Best regards,
> > >
> > > The Authors

---

### Official Review · Reviewer_LbRf · 2024-11-04

**Soundness:** 3
**Presentation:** 3
**Contribution:** 3
**Rating:** 6
**Confidence:** 4

**Summary:**

The manuscript introduces a way to approximate a non-linear PDE operator with a GNN block completed with a Laplace operator block.
The examples include a varity of linear and linear PDEs.

**Strengths:**

The way to approximate a non-linear PDE operator with a GNN block completed with a Laplace operator block looks fairly original.

**Weaknesses:**

1.	The ultimate goal of the manuscript is to learn the nonlinear operator $F$.  However, it is not explained why it is needed while $F$ is known and given.
2.	The way how the authors approximate the nonlinear operator F, Eq. (6), evidently has some limitations. For example, it won’t work well when $F = (\Delta u)^2 $ and so on.  Consider being more specific in defing the operator F in Eq. (1).
3.	Section 4 is difficult to follow since the actual PDEs and their coefficients are not specified.
4.	It is not really explained why paddling is needed to incorporate boundary conditions.

**Questions:**

1.	Could you give mathematical and physical interpretation of $z_i$, Eq. (5)? What activation functions are used to parametrize $z_i$? How do you formally differentiate between coarse and fine meshes?
2.	Section 3.4 implies that over all there are $T/M$ loss functions. Are these loss values combined together somehow? What is ground truth data in this setting?
3.	Could you include experiments with $F = \text{div} (D \nabla u)$ with either $D(x,y)=1 $ or $D(x,y)=1+4x+5y$ (or other varying from 1 to 10)? It will give to the reader a good understanding of approximation properties of Eq.(6).
4.	The discussion in the manuscript covers only 2D spatial data. Does the proposed method admit a generalization to the 3D case? How the Laplace block will like in this case?
5.	The RK2 and RK4 schemes are known to be applicable to non-stiff problems only due to stability limitations. What scheme would you suggest for a stiff PDE of form Eq. (1) and how would you implement such a scheme?

---

> ### Author Response · Authors · 2024-11-21
> **Reply to Reviewer LbRf (Part 1)**
>
> We sincerely thank you for the constructive comments and suggestions, which have been carefully considered and incorporated into the revised version of our paper.
>
> > **Q1. The nonlinear operator $F$ in Eq.(1) is only partially known and the limitations of Eq. (6) to approximate it.**
>
> **Reply:** Excellent comment! In all experiments in this paper, $F$ is **not** completely known and given. As mentioned in Section 3.1 (page 3), considering that many physical phenomena involve diffusion processes, we assume the diffusion term in the PDE is known as a *priori* knowledge, which means $F$ is only partially known.
>
> Therefore, we design the Laplace block to learn the increment caused by the diffusion term in the PDE and the GNN block aims to learn the increment induced by other unknown mechanisms or sources. Considering that the diffusion term present in most PDE systems (such as reaction-diffusion systems and the NS equations) is typically linear, the output of the Laplace block and the GNN block are directly combined through addition in Eq. (6). We acknowledge that this approach cannot be applied to other nonlinear or high-order diffusion terms (e.g., $\nabla^4$), which is a limitation that has been clarified in Section 5 (page 10) in the revised paper. However, if the form of the diffusion term is already known, e.g., $(\Delta u)^2$, our approach remains applicable by using the squared Laplace block.
>
> > **Q2. The actual PDEs and their coefficients in Section 4.**
>
> **Reply:** Due to the page limitation, the details of the actual PDEs and their coefficients are provided in Appendix D.2 (page 21). For your reference, we present the actual PDEs and their coefficients again below.
>
> - **Burgers' equation:** $\boldsymbol{\dot{u}} = \nu \Delta \boldsymbol{u} - \boldsymbol{u} \cdot \nabla \boldsymbol{u}$, where the diffusion coefficient $\nu$ is $5\times10^{-3}$.
>
> - **FN equation:** $\dot{u}=\mu_u\Delta u + u - u^3 - v + \alpha, \ \dot{v}=\mu_v\Delta v + (u-v)\beta$, where the diffusion coefficients $\mu_u$ and $\mu_v$ are respectively $1$ and $10$ and the reaction coefficients $\alpha$ and $\beta$ are respectively $0.01$ and $0.25$.
>
> - **GS equation:** $\dot{u}=\mu_u\Delta u - uv^2 + \mathcal{F}(1-u), \  \dot{v}=\mu_v\Delta v + uv^2 - (\mathcal{F} + \kappa)v$, where the diffusion coefficients $\mu_u$ and $\mu_v$ are respectively $2.0\times10^{−5}$ and $5.0\times10^{−6}$ and the reaction coefficients $\mathcal{F}$ and $\kappa$ are respectively $0.04$ and $0.06$.
>
> - **Cylinder flow:** $\boldsymbol{\dot u} = - \boldsymbol{u} \cdot \nabla \boldsymbol u - \frac{1}{\rho} \Delta p + \frac{\mu}{\rho} \Delta \boldsymbol{u} + \boldsymbol{f}$, where the fluid density $\rho$ is $1$, the fluid viscosity $\mu$ is $5.0\times10^{-3}$, the external force $\boldsymbol{f}$ is $0$, and the cylinder diamater $D=2$.
>
> > **Q3. Why padding is needed to incorporate boundary conditions.**
>
> **Reply**: Great question! As mentioned in Section 3.3 (page 6), the solution of spatiotemporal PDEs is determined by the governing equations, initial conditions (ICs), and boundary conditions (BCs). Although our model is also applicable without BCs, it is of common practice to have the *a priori* knowledge of BCs for a given spatiotemporal dynamical system. Nevertheless, how to effectively leverage BCs to improve the model's performance remains an open challenge. Some existing approaches impose BC as soft constraints in loss functions (such as PINNs [1]) or entirely rely on neural networks to learn patterns that satisfy BCs (such as MGN [2], MP-PDE [3]), which inevitably leads to errors accumulating in long-term roll-out prediction. In contrast, our proposed padding strategy enforces strict compliance with BCs (once given), helping constrain the solution space and accurately simulate physical systems for long term.
>
> ***References:***
>
> [1] Raissi et al., JCP 2019.
>
> [2] Pfaff et al., ICLR 2021.
>
> [3] Brandstetter et al., ICLR 2022.

---

> > ### Comment · Reviewer_LbRf · 2024-11-21
> >
> > Dear authors,
> >
> > Thank you for your comments.
> > My remarks were addressed.

---

> > > ### Author Response · Authors · 2024-11-22
> > > **Thank you for your feedback**
> > >
> > > Dear Reviewer LbRf,
> > >
> > > Thanks for your prompt feedback. Since your concerns have been addressed, we appreciate very much if you would consider to increase the score of our paper (which still remains a negative rating).
> > >
> > > Thank you very much!
> > >
> > > Best regards,
> > >
> > > The Authors

---

> ### Author Response · Authors · 2024-11-21
> **Reply to Reviewer LbRf (Part 2)**
>
> > **Q4. Mathematical and physical interpretation of $z_i$ in Eq. (5) and how to differentiate between coarse and fine meshes.**
>
> **Reply:** Thanks for your question. As mentioned in Section 3.2.2 (page 5), Mesh Laplace (e.g., the discrete Laplace-Beltrami operator) exhibits good accuracy on fine meshes but yields unsatisfactory results on coarse meshes. To address this issue, we employ a lightweight neural network to rectify the Laplacian estimation on coarse meshes. The quantity $z_i$ in Eq. (5) denotes the output of the lightweight network, which can be computed as $z_i = \text{MLP} _{\beta} \ \circ \text{MPNNs} \ \circ \text{MLP} _{\alpha} (u_i, x_i, u_j - u_i, x_j - x_i, ...)$. The motivation of Eq. (5) is inspired by the predictor-corrector framework, such as the Euler method with the trapezoidal rule or adaptive gradient algorithms, where an initial rough prediction is made and then refined.
>
> The key distinction between coarse and fine meshes lies in the number of discrete nodes within the same domain, which significantly affects the accuracy of discrete operator approximations. As stated in Section 4.1 (page 7), the MSE of Mesh Laplace on a coarse mesh with $N=983$ is $2574$. In contrast, we complement this experiment and obtain the result that the MSE of Mesh Laplace on the fine mesh with $N=16641$ $(129\times129)$ is only $3.66$ (see **Table A** below).
>
> **Table A: The MSE of Mesh Laplace on different resolutions.**
> | Resolutions (N) | 983  | 16641 (129x129) |
> | -------- | -------- | -------- |
> | MSE     | 2574     |   3.66   |
>
>
> > **Q5. The $T/M$ loss functions in Section 3.4 and the ground truth data.**
>
> **Reply:** There seems to be some misunderstanding here. Section 3.4 (page 6) does not involve $T/M$ loss functions. As explained in Section 3.4, due to GPU memory limitations, directly training the model on the entire long time series ($T$ steps, $\boldsymbol{u}^0, \dots, \boldsymbol{u}^{T-1}$) is impractical. To address this, we segment the time series into multiple shorter time sequences of $M$ steps ($T \gg M$), which can be represented as $\boldsymbol{u}^{s_0}, \dots, \boldsymbol{u}^{s_{M-1}}$. Within the input of $\boldsymbol{u}^{s_0}$, the model rolls out for $M-1$ steps to generate predictions $\boldsymbol{\hat{u}}^{s_1}, \dots, \boldsymbol{\hat{u}}^{s_{M-1}}$, computes the segment loss against the ground truth, and backpropagates, rather than summing $T/M$ losses. These segments collectively form the entire training dataset, with each segment functioning as a batch to some extent. The clarification have been included in Section 3.4 (page 6) in the revised paper.
>
> As mentioned in Appendix D.2 (page 21), we use COMSOL, a multiphysics simulation software based on numerical methods, to generate trajectories ($T$ steps) with different ICs for training and testing, which serve as our ground truth data.
>
> > **Q6. Experiments on $F=\text{div}(D\nabla \boldsymbol{u})$ with $D(x,y)=1+4x+5y$.**
>
> **Reply:** Excellent suggestion! We took your suggestion and generated 10 trajectories with 1600 time steps for training and another 10 trajectories for testing using COMSOL, based on your suggested equation $F=\text{div}(D\nabla \boldsymbol{u})$ with $D(x,y)=1+4x+5y$. We trained our model and obtained the testing MSE of $4.69\times 10^{-5}$ (sufficiently small). The results further demonstrate the efficacy of our proposed model.
>
> > **Q7. Generalization to the 3D Case.**
>
> **Reply:** Insightful question! The Mesh Laplace module in our Laplace block is a discrete differential geometry operator designed for triangulated 2-manifolds [4]. This makes it applicable to 2D scenarios and 3D surfaces, but not to 3D volumes. For 3D PDE systems, alternative methods (such as cubic FEM) to compute the Laplacian on irregular domains need to be incorporated into the models, which sets forward our future work. Thank you for your great suggestion!
>
> ***Reference:***
>
> [4] Meyer et al.,  Visualization and Mathematics III 2003.

---

> ### Author Response · Authors · 2024-11-21
> **Reply to Reviewer LbRf (Part 3)**
>
> > **Q8. The stability limitations of RK2 and RK4 schemes for stiff problems.**
>
> **Reply:** Great comment! We agree on the stability limitations of RK2 and RK4 schemes for solving stiff PDEs, which our model may also suffer from. Explicit methods typically require a small time-stepping $\delta t$ to ensure numerical stability compared to implicit methods. However, our model demonstrates more relaxed time-stepping $\delta t$ requirements. We have compared the stability of PhyMPGN with both explicit and implicit COMSOL simulations, and observed that our model can generalize to larger $\delta t$ unseen during training.
>
> Specifically, our model was trained with $\delta t=0.001$ but successfully tested with $\delta t=0.004, 0.008$ and even $0.040$ without retraining. The stability and MSE of three methods on increasing $\delta t$ are presented in **Table B**. While we acknowledge using implicit methods in models offer better stability than explicit methods, but how to integrating them into learning models is beyond the scope of this paper. However, this set path for our future research.
>
> **Table B: The stability and MSE of three methods on increasing $\delta t$.**
> | dt | COMSOL (RK4) | COMSOL (implicit) | PhyMPGN |
> | -------- | -------- | -------- | -------- |
> | 0.001     | stable (4.91e-5)     | stable (4.91e-5)     | stable (2.99e-4) |
> | 0.004     | stable (5.33e-5)     | stable (5.22e-5)     | stable (2.99e-4) |
> | 0.008     | unstable (non-convergence) | stable (6.49e-5) | stable (2.98e-4) |
> | 0.040     | unstable (non-convergence)     | stable (3.72e-4) | stable (3.77e-4) |
> | 0.050     | unstable (non-convergence)     | unstable (use smaller $\delta t$ automatically)     | unstable (3.72e+0)|
>
>
> In summary, we appreciate your constructive comments and suggestions. Please let us know if you have any other questions. We look forward to your feedback!

---

> ### Author Response · Authors · 2024-11-25
> **Request your feedback before the end of the discussion period**
>
> Dear Reviewer LbRf,
>
> We sincerely appreciate your thoughtful and constructive comments. As the author-reviewer discussion period will end soon, please feel free to share any additional questions or comments.  We will do our best to address them.
>
> Thank you once again for your time and valuable feedback. Looking forward to your response!
>
> Sincerely,
>
> The Authors

---

### Official Review · Reviewer_cFxe · 2024-11-04

**Soundness:** 3
**Presentation:** 3
**Contribution:** 3
**Rating:** 8
**Confidence:** 5

**Summary:**

The paper is related to PDE systems solution in two-dimensional space on irregular domains. The problem is addressed with the mix of GNNs and classical numerical schemes. The authors propose several novel elements, such as trainable Laplace block, second order Runge-Kutta scheme for simulating temporal dynamics and a padding strategy to incorporate boundary conditions of different types. These improvements allowed the authors to achieve new quality of simulation. The obtained model has good generalization capabilities over initial conditions and over the unseen types of flow (such as another Reynolds number)

**Strengths:**

* The authors have shown that spatiotemporal dynamics can be modeled by GNN in spatial domain together with 2nd order numerical scheme in temporal domain
* The authors proposed and tested a combination of reasonable improvements to the previous works with similar techniques (Pfaff et al. 2021). There key additions are Laplace block and padding strategy for different types of boundary conditions
* The proposed improvements allowed authors to outperform the competitors in terms of quality for chosen test problems
* Generalization and ablation studies substantiate the authors contributions

**Weaknesses:**

The limitations of the work were not addressed properly. In particular:
* applicability to the PDE systems without the Laplace operator
* simulation for the big number of time steps (for example this occurs for weather simulations)
* boundary conditions padding for the domain with non-straight boundary – it’s not clear how to work with them

Also the experiments are insufficient:
* It’s important to investigate the computational requirements of the proposed approach for training and inference in dependence on the important parameters (number of time steps, size of the training dataset, etc.) in order to assess its applicability in practice
* Also it would be useful to understand how the segment size M in 3.4 influences the results and computational requirements

Also the paper is hard to read and it needs some restructurization. Probably  the introduction can be shortened a little bit together with the beginning of the abstract and authors could explain in more details how the model is trained and how different types of boundary conditions are introduced for curved boundary

**Questions:**

* Why in eq. (7) for loss function you use only first and last step? What will change if we will compare full rollouts?
* How the padding strategies will work for boundary conditions on irregular shapes?
* What do you mean when you say that boundary conditions are used as a priori physics knowledge?
* Will the Laplace block still be useful for PDEs without the Laplace operator?

also look for the suggestions arising from weaknesses

---

> ### Author Response · Authors · 2024-11-21
> **Reply to Reviewer cFxe (Part 1)**
>
> Thank you for your constructive comments and suggestions, which have been carefully considered and incorporated into the revised version of our paper.
>
> > **Q1. Applicability to the PDE systems without the Laplace operator.**
>
> **Reply:** Insightful suggestion! As mentioned in Section 3.2 (page 3), the Laplace block in our model is specifically designed to learn the diffusion process while the GNN block is tasked with learning the dynamics governed other unknown mechanisms or sources. In cases where the dynamical system does not involve diffusion (aka, PDE systems without the Laplace operator), the Laplace block can be removed from the model architecture. However, the boundary conditions (BCs) can still be effectively encoded using our padding strategy, which helps to constrain the solution space and accurately predict spatiotemporal dynamics for a long-term horizon.
>
> > **Q2. Simulation for the big number of time steps like weather simulations.**
>
> **Reply:** Great comment! Compared to some well-known neural-based models for PDE systems, the number of time steps predicted by our model is much larger. For instance, FNO [1] predicts 50 steps, MGN [2] predicts 600 steps, and MP-PDE [3] predicts 250 steps. In contrast, our model predicts 1600 steps for Burgers' equation and 2000 steps for the other three equations. Additionally, we evaluate the extrapolation capability of our model on the cylinder flow problem, achieving predictions for up to 4000 steps (see lines 485-487 in the revised paper).
>
> To our best knowledge, neural-based models for weather forecasting usually predict for a small number of time steps but a big time interval. For instance, DGMR [4] predict 16 steps, GraphCast [5] predict 40 steps, NowcastNet [6] predict 18 steps.
>
> > **Q3. Boundary conditions are used as a priori knowledge.**
>
> **Reply:** Good question! That means that the boundary conditions (BCs) are known, and we use the padding strategy to encode these known BCs into our model. Below are several examples of BCs' formulas:
>
> - a Dirichlet BC can be expressed as ${u}(\boldsymbol{x}) = 0, \boldsymbol{x} \in \Gamma_d$, where $\Gamma_d$ represents the Dirichlet boundary of domain.
> - a Neumann BC can be expressed as $\frac{\partial {u}(\boldsymbol x)}{\partial \boldsymbol{n}} = 0, \boldsymbol{x} \in \Gamma_n$, where $\Gamma_n$ represents the Neumman boundary.
> - a Robin BC can be expressed as $0.1 {u}(\boldsymbol{x}) + 0.2\frac{\partial {u}(\boldsymbol x)}{\partial \boldsymbol{n}} = 1, \boldsymbol{x} \in \Gamma_r$, where $\Gamma_r$ represents the Robin boundary.
> - a periodic BC is expressed as ${u}(\boldsymbol{x}_1) = {u}(\boldsymbol{x}_2), \boldsymbol{x}_1 \in \Gamma _{p1},\boldsymbol{x}_2 \in \Gamma _{p2}$, where $\Gamma _{p1}$ and $\Gamma _{p2}$ are periodic boundary for each other.
>
> The general formulas for these four types of boundary conditions are summarized in Appendix Table A.1 (page 20) in the revised paper.
>
> > **Q4. How the padding strategy works for BCs on irregular shapes.**
>
> **Reply:** Good question! The padding strategy is discussed in Section 3.3 (page 6). For Dirichlet BCs, nodes on the Dirichlet boundary are directly assigned specific values. For Neumann/Robin BCs, ghost nodes are created symmetrically with respect to the nodes near the boundary (along the normal direction of the boundary), and their padded values depends on derivatives in the normal direction. The goal is to ensure the true nodes, the boundary, and the ghost nodes satisfy the BCs in a manner similar to the central difference method. For periodic BCs, nodes near one boundary $\Gamma _{p1}$ are flipped and placed near the corresponding boundary $\Gamma _{p2}$, achieving a cyclic effect during message passing. The clarification have been included in Section 3.3 (page 6) and Appendix C (page 18) in the revised paper.
>
> Details of the corresponding padding formulas along with a diagram illustrating the padding strategy are provided in Appendix C (page 18) and Appendix Figure A.2 (page 19). Full source code and implementation details will also be made available after the peer review.
>
> ***References:***
>
> [1] Li et al., ICLR 2021.
>
> [2] Pfaff et al., ICLR 2021.
>
> [3] Brandstetter et al., ICLR 2022.
>
> [4] Ravuri et al., Nature 2021.
>
> [5] Lam et al., Science 2023.
>
> [6] Zhang et al., Nature 2023.

---

> ### Author Response · Authors · 2024-11-21
> **Reply to Reviewer cFxe (Part 2)**
>
> > **Q5. Add computational cost of training and inference in dependence on the important parameters.**
>
> **Reply:** Good suggestion! First, we would like to clarify that the computational cost of our Laplace block and padding strategy is minimal.The computation of the Mesh Laplace in the Laplace block includes only matrix multiplication, while the cost of the padding strategy involves merely copying the corresponding nodes in the domain which is negligible.
>
> The training time and testing time primarily depends on factors such as the number of trainable parameters, the types of numerical integrators used, the size of the dataset (including the number of time steps and trajectories), and the batch size. For instance, in the cylinder flow case, which includes 4 trajectories with 2000 time steps and approximately 1600 nodes in the domain for training, the computational costs of models with around 950k parameters using different numerical integrators are summarized in **Table A** below.
>
> We trained our model for 1600 epochs using four NVIDIA RTX 4090 GPUs and conducted inference for 2000 time steps per trajectory on a single GPU. The model demonstrates fast inference speeds, with times under one minute. Leveraging the parallel processing capabilities of GPUs, the inference speed can be further improved by processing multiple trajectories in batches. The above content have been included in Section 4.2 (page 10), Appendix E (page 24-26), and Appendix Table A.11 (page 26) in the revised paper.
>
> **Table A: The training and inference time of our models employing different types of numerical integrators.**
> |  | Training time  | Inference time (per trajectory) | Inference time (batch, 5 trajectories) |
> | -------- | -------- | -------- | -------- |
> | PhyMPGN (Euler)    | 5.2 h     | 8.98 s     | 13.47 s     |
> | PhyMPGN (RK2)| 14.4 h    | 17.57 s     | 27.08 s     |
> | PhyMPGN (RK4)| 39.3 h     | 34.99 s     | 53.49 s     |
>
> > **Q6. Impact on the results and computational requirements of the segment size $M$.**
>
> **Reply:** Great comment! Follow your suggestion, we conducted experiments using the cylinder flow dataset in the ablation study. We trained our model with segment sizes $M=10, 15, 20$, and the MSEs on the testing sets and the training time are presented in **Table B** below. It shows that the bigger the segment size $M$, the better the model's prediction accuracy.
>
> Intuitively, longer time segments can be advantageous for capturing long-term dependencies, thereby improving the model's long-term prediction capabilities. However, excessively long segments may introduce challenges during training, such as increased memory consumption and difficulties in achieving convergence. Thus, we choose $M=20$ to strike a balance.
>
> Additionally, since we need to adjust the batch size to fit the GPU memory as the segment size $M$ varies, the training time does not scale linearly with $M$. Specifically, the batch sizes for $M=15$ and $M=20$ are the same, while the batch size for $M=10$ is double that of the other two configurations. Therefore, the training time differences are also reasonable.
>
> The additional results and discussions have been included in Section 4.3 (page 10), Appendix E (page 26), and Appendix Table A.12 (page 26) in the revised paper.
>
> **Table B: The MSE and the training time with different segment size $M$.**
> | $M$ | $Re=[200, 280, 360]$ | $Re=[440, 480, 520]$ | $Re=[600, 800, 1000]$ | Training time |
> | -------- | -------- | -------- | -------- | -------- |
> | 10    | 2.43e-3     | 1.48e-1     | 2.42e-1     | 8.3 h |
> | 15    | 9.34e-4     | 3.05e-2     | 1.34e-1     | 15.0 h |
> | 20    | 2.14e-4     | 3.56e-2     | 1.25e-1     | 14.4 h |
>
>
> > **Q7. Loss function at only first and last steps vs. full rollouts.**
>
> **Reply:** Great question! The idea of backpropagating only at the first and last steps is indeed inspired by the pushforward trick introduced in MP-PDE [3]. As for full rollouts, intuitively, due to the nature of rollouts, backpropagating the loss behind the first time step causes the information from the first step to be traced back as well. This means that using the full rollout loss can significantly enhance short-term forecasting capabilities, but it may also lead to overfitting and be less effective for long-term predictions. On the other hand, backpropagating the loss only at the first and last steps strikes a balance, allowing the model to learn both short-term and long-term dynamics while reducing computational cost.
>
> In summary, we appreciate your constructive comments and suggestions. Please let us know if you have any other questions. We look forward to your feedback!

---

> ### Author Response · Authors · 2024-11-23
> **Looking forward to your feedback**
>
> Dear Reviewer cFxe,
>
> Again, thanks for your constructive comments. We would like to follow up on our rebuttal to ensure that all concerns have been adequately addressed. If there are any further questions or points that need discussion, we will be happy to address them. Your feedback is invaluable in helping us improve our work, and we eagerly await your response.
>
> Thank you very much for your time and consideration.
>
> Best regards,
>
> The Authors

---

> ### Author Response · Authors · 2024-11-25
> **Request your feedback before the end of the discussion period**
>
> Dear Reviewer cFxe:
>
> As the author-reviewer discussion period will end soon, we would appreciate it if you could review our responses at your earliest convenience. If there are any further questions or comments, we will do our best to address them before the discussion period ends.
>
> Thank you very much for your time and efforts. Looking forward to your response!
>
> Sincerely,
>
> The Authors

---

> ### Comment · Reviewer_cFxe · 2024-11-27
> **Reply to replies and score increase**
>
> Firstly, thanks to the authors for such a rigorous work on all the comments!
>
> The authors have addressed all of my questions both in revised version of the paper and in the answers, so I decided to increase:
> * Presentation 2->3
> * Rating 6->8
>
> The computational requirements for the final solution are rather high so I feel it would be difficult to find practical applications to it (compared to Comsol, for example, which doesn't require time for training). However I believe that the proposed solution will benefit the physics simulation part of the ML society and has enough room for future improvements to achieve better practical efficiency for real world cases. Hope to have an interesting discussion with the authors at the conference!

---

> > ### Author Response · Authors · 2024-11-27
> > **Thank you for your feedback**
> >
> > Dear Reviewer cFxe,
> >
> > Thank you very much for your recognition and positive feedback. We greatly value the time and effort you dedicated to reviewing our paper. We are also delighted by your interest in discussing our work further and look forward to the opportunity for an in-depth exchange at the conference.
> >
> > Best regards,
> >
> > The Authors

---

### Official Review · Reviewer_QKXd · 2024-11-04

**Soundness:** 3
**Presentation:** 3
**Contribution:** 3
**Rating:** 8
**Confidence:** 4

**Summary:**

The work focuses on solving PDE with low-resolution unstructured meshes. In particular, the authors incorporate the discrete Laplace–Beltrami operator to express the diffusion process and a specific padding method to satisfy boundary conditions.

The experimental evaluation demonstrates that the introduced Laplace block (GNN + Laplace Beltrami operator) is strongly expressible to approximate Laplacian compared to the raw numerical scheme and machine learning models considered in the evaluation. In addition, the method has a stable prediction for long-time series prediction with generalization regarding initial condition, boundary condition, and Reynolds number to some extent.

**Strengths:**

* The numerical experiments are thorough, and the proposed method demonstrates its high performance in various tasks and generalization scenarios. In particular, the method can extrapolate the Reynolds number to some extent, which is impressive.
* The method is simple and easy to understand. Due to its simplicity, the reviewer expects the method to be compatible with other GNN-based methods, which would be solid knowledge to the community.

**Weaknesses:**

* The novelty of the proposed method is limited. The connection between GNN and the Laplace (–Beltrami) operator is not new (e.g., baselines in Section 4.1 and [Li et al. AAAI 2018 https://arxiv.org/abs/1801.07606 ]). Since the performance is better than the baseline, there might be a big improvement for the community, but it is unclear what it is. The author could have elaborated on what were the problems of existing methods and how they overcame the difficulties.
* The method to treat boundary conditions in unstructured meshes is also not new and proposed in [Horie et al. NeurIPS 2022 https://openreview.net/forum?id=B3TOg-YCtzo ]. At least the author should mention existing works and possibly compare the proposed method with them to discuss the pros and cons.
* For time-series training, the motivation to incorporate the first step in the loss (Eq 7). As mentioned in the paper, [Brandstetter et al. 2022] proposed a method called the pushforward trick, which computes the loss using only the last time step, demonstrating the trick performs better than adding noise. In the present work, Eq 7 seems to contradict the previous works. Thus, the authors should clarify the reason why they have the first step with noise and demonstrate the proposed one is better.

**Questions:**

* The method seems quite simple yet powerful, as seen in the numerical experiments. However, the reviewer wonders where the expressibility and generalizability come from. Could the authors explain the reason for high performance, connecting with the method proposed in the paper?
* Is the method generalizable regarding mesh resolution? For instance, learning on coarse meshes and predicting on finer meshes or vice versa would be interesting to see, while that would be out-of-scope of the work.

---

> ### Author Response · Authors · 2024-11-21
> **Reply to Reviewer QKXd (Part 1)**
>
> Thank you for your constructive comments and suggestions! We have carefully addressed them, and the following responses have been incorporated into the revised paper.
>
> >**Q1. Connection of the Laplace operator and padding strategy for BCs with previous methods; the novelty, expressibility and generalizability.**
>
> **Reply:** Good question! We would like to clarify the connection between our contributions with several previous methods one by one below:
>
> ***Connection of the Laplace operators with [1]:*** The laplace operators mentioned in [1] (Li et al.) refer to as the graph Laplacian, namely, $L := D - A$, where $A$ is the adjacency matrix and $D$ is the degree matrix. However, the Laplace-Beltrami operator [2] is defined as $\Delta f = \text{div}(\text{grad} \ f)$, where $f$ is a real-value function defined on a differentiable manifold, and $\text{grad}$ and $\text{div}$ are the gradient and divergence on the manifold, respectively. The corresponding discrete Laplace-Beltrami operators used in our paper are often represented as $\Delta f_i = \frac{1}{d_i}\sum _{j \in \mathcal{N} _i} w _{ij} (f_i - f_j)$. In summary, the graph Laplacian $L$ and the Laplace-Beltrami operator in our paper are completely different.
>
> ***Connection of the Laplace operator with SDL [3]:*** Seo et al. proposed a spatial difference layer (SDL) to define learnable Laplace operator, namely, $(^w \Delta f)_i = \sum _{j: (i, j) \in \mathbb{E}} w^{(l_1)} _{ij} (f_i - w^{(l_2)} _{ij} f_j)$, where $w _{ij}$ is the output from GNN. SDL's definition is totally different from our Laplace block, which reads $\Delta f_i = \frac{1}{d_i}(z_i + \sum _{j \in \mathcal{N}_i} w _{ij} (f_i - f_j))$, where $z_i$ is the output from GNN, $d_i$ and $w _{ij}$ is a fix value related to geometry shape. Note that the discrete Laplace-Beltrami operator in our Laplace block preserves much geometry information of coarse meshes and offers a more feasible inductive bias to learn the diffusion term. In contrast, SDL entirely replies on neural networks to learn the diffusion pattern. As demonstrated in Section 4.1 (page 7), the approximation ability of our Laplace block significantly outperforms SDL.
>
> ***Connection of treating BCs on unstructured meshes with [4]:*** Both PENN [4] (Horie et al.) and our method encode boundary conditions (BCs). However, there are several key differences between the two approaches, namely,
>
> - *Methods:* PENN designs the special neural layers and modules for the model to satisfy BCs, while we develop a padding strategy directly applied to the features.
> - *Types of BCs:* PENN proposes a DirichletLayer and pseudoinverse decoder for Dirichlet BCs, and NeumannIsoGCN for Neumann BCs. In contrast, our padding strategy can be applied to four types of BCs (Dirichlet, Neumman, Robin, periodic), which can be seamlessly integrated into any GNN model and thus has a better generality.
> - *Accuracy:* Both PENN and our padding strategy are designed to completely satisfy Dirichlet BCs without any error. However, neither method can enforce other BCs without error.
> - *Efficiency:* During training, the additional cost for PENN to satisfy BCs involves constructing the pseudoinverse decoder after the parameters are updated. However, the cost of our padding strategy involves merely copying the corresponding nodes in the domain and is essentially negligible.
>
> ***Novelty, expressibility and generalizability:*** In summary, our Laplace block and BC padding strategy differ significantly from the aforementioned methods, representing the novelty and key contributions of our paper. The learnable Laplace block, which encodes the discrete Laplace-Beltrami operator, along with the padding strategy for BCs both aid and guide the GNN learning in a physically feasible solution space. This is where our model's expressibility and generalizability stem from.
>
> We have added the above comparison in Section 3.3 (page 6) and Appendix C (page 19) in the revised paper.
>
> ***References:***
>
> [1] Li et al., AAAI 2018.
>
> [2] Reuter et al., Computer & Graphics 2009.
>
> [3] Seo et al., ICLR 2020
>
> [4] Horie et al. NeurIPS 2022

---

> ### Author Response · Authors · 2024-11-21
> **Reply to Reviewer QKXd (Part 2)**
>
> > **Q2. Clarification the motivation of the loss (Eq. 7) and its connection with MP-PDE (Branstetter et al.).**
>
> **Reply:** Great comment! Several previous approaches [5, 6, 7, 8] for modeling physics dynamics have demonstrated the effectiveness of introducing training noise to alleviate issues such as distribution shift and error accumulation. To achieve a similar goal, MP-PDE's pushforward trick [9] only backpropagates errors on the last unroll step. However, MP-PDE does not claim that this trick performs better than adding noise. In contrast, they consider both the pushforward trick and the introduction of training noise as adversarial losses.
>
> Our training method is largely inspired by previous approaches and does not contradict them. While MP-PDE only rolls out for 2 steps, our model typically rolls out for more steps (e.g., $M=20$). In such cases, computing the loss solely at the last prediction could lead to a lack of guidance on intermediate states. To address this, we backpropagate at both the first and last predictions.
>
> As mentioned in Section 3.4 (page 6), the long time series ($T$ steps, $\boldsymbol{u}^0, \dots, \boldsymbol{u}^{T-1}$) are segmented into multiple shorter time sequences of $M$ steps ($T \gg M$), where each segment is represented as $\boldsymbol{u}^{s_0}, \dots, \boldsymbol{u}^{s_{M-1}}$. We introduce a small noise to the first frame $\boldsymbol{u}^{s_0}$ as the input for the model in each segment. With the input $\boldsymbol{u}^{s_0}$, the model rolls out for $M-1$ steps to generate predictions ($\hat{\boldsymbol{u}}^{s_1}, \dots, \hat{\boldsymbol{u}}^{s_{M-1}}$) but backpropagation is only applied to the first and last predictions. Therefore, the loss function is defined as $\mathcal{L}=\text{MSE}(\boldsymbol{u}^{s_1}, \hat{\boldsymbol{u}}^{s_1}) + \text{MSE}(\boldsymbol{u}^{s_{M-1}}, \hat{\boldsymbol{u}}^{s_{M-1}})$.
>
> We apologize for any confusion caused by the statement in our orignal paper. We have provided the clarification in Section 3.4 (page 6) in the revised paper.
>
> > **Q3. Generalization on mesh resolution.**
>
> **Reply:** Thanks for your question. When the model trained on coarse meshes is used to predict on finer meshes, we observe that the MSE increases by orders of magnitude. This indicates that our model does not generalize well across different mesh resolutions. This issue likely arises because the learnable Laplace block only learn how to correct the results on the coarse mesh so fail on unseen finer mesh. As you said, it's out of the scope of our paper.
>
> In summary, we appreciate your constructive comments and suggestions. Please let us know if you have any other questions. We look forward to your feedback!
>
> ***References:***
>
> [5] Sanchez-Gonzalez et al., ICML 2018
>
> [6] Sanchez-Gonzalez et al., ICML 2020
>
> [7] Pfaff et al., ICLR 2021
>
> [8] Stachenfeld et al., ICLR 2022
>
> [9] Brandstetter et al., ICLR 2022

---

> ### Author Response · Authors · 2024-11-23
> **Looking forward to your feedback**
>
> Dear Reviewer QKXd,
>
> Again, thanks for your constructive comments. We would like to follow up on our rebuttal to ensure that all concerns have been adequately addressed. If there are any further questions or points that need discussion, we will be happy to address them. Your feedback is invaluable in helping us improve our work, and we eagerly await your response.
>
> Thank you very much for your time and consideration.
>
> Best regards,
>
> The Authors

---

> ### Author Response · Authors · 2024-11-25
> **Request your feedback before the end of the discussion period**
>
> Dear Reviewer QKXd:
>
> As the author-reviewer discussion period will end soon, we would appreciate it if you could review our responses at your earliest convenience. If there are any further questions or comments, we will do our best to address them before the discussion period ends.
>
> Thank you very much for your time and efforts. Looking forward to your response!
>
> Sincerely,
>
> The Authors

---

> > ### Comment · Reviewer_QKXd · 2024-11-26
> >
> > Thank you for the detailed rebuttal. My concerns are almost addressed. One remaining point would be a comparison between PENN in experiments. It is because the efficiency stated by the authors is somehow qualitative, and quantitative comparison brings more information, given that the proposed method uses padding, which may also increase computation cost. In particular, PENN may be able to eliminate the cost of the pseudoinverse computation at the prediction phase, but the proposed method essentially has an additional cost coming from padding, even at the prediction phase. If my statement above is true, I recommend including the drawback of the padding method to clarify the limitation of the method.
> >
> > Nevertheless, I acknowledge the authors' contributions to the ML community. Therefore, I updated the score accordingly as follows:
> > * Soundness: 2 -> 3
> > * Rating: 6 -> 8

---

> > > ### Author Response · Authors · 2024-11-26
> > > **Reply to the additional question**
> > >
> > > We sincerely appreciate your recognition and the positive feedback of our work.
> > >
> > > > **Question: Quantitative experiments about the padding strategy.**
> > >
> > > **Reply:** Thanks again for your valuable suggestion. We acknowledge that our padding strategy leads to some extra computational cost during both training and testing, whereas PENN introduces an approach to encode BCs that enables to eliminate the cost of the pseudoinverse computation during testing.
> > >
> > > However, as previously mentioned, the cost of our padding strategy involves merely copying the corresponding nodes in the domain, resulting marginal increase of the computational cost. To quantify this time overhead, we conducted an experiment comparing the inference time of our models with and without the padding strategy for predictions over 1600 time steps, as shown in **Table A** below. The results demonstrate that the overhead introduced by the padding strategy is small.
> > >
> > > The clarification and the quantitative comparison have been included in **Appendix C (page 19)** in the revised paper.
> > >
> > > **Table A: The inference time of models with and without the padding strategy.**
> > > | Model | Inference time (s) |
> > > | -------- | -------- |
> > > | PhyMPGN (padding)     | 10.71     |
> > > | PhyMPGN (without padding)     | 10.02     |

---

> > > > ### Comment · Reviewer_QKXd · 2024-11-26
> > > >
> > > > Thank you for the prompt response. Now everything is clear. Great work!

---

> > > > > ### Author Response · Authors · 2024-11-26
> > > > > **Thank you for your feedback**
> > > > >
> > > > > Dear Reviewer QKXd,
> > > > >
> > > > > Thank you very much for your positive feedback. Your time and effort placed on reviewing our paper are highly appreciated!
> > > > >
> > > > > Best regards,
> > > > >
> > > > > The Authors

---

### Author Response · Authors · 2024-11-21
**General reply**

Dear Reviewers:

We sincerely thank you for your valuable and constructive feedback, which has been instrumental in enhancing our paper.  We have posted the point-to-point reply to each question/comment raised by you.

In addition, comprehensive revisions and adjustments (indicated in red color) have been made in the revised paper (please see **the updated .pdf file**), which include:
- Comparing our padding strategy with the PENN method (Section 3.3 of page 6 and Appendix C of page 19).
- Clarifying the model training process and the loss function (Section 3.4 of page 6).
- Refining the statement of the padding strategy (Section 3.3 of page 6).
- Analyzing computational efficiency, including training and inference times (Section 4.2 of page 10 and Appendix E of page 24).
- Clarifying the statement about the time-stepping requirements of implicit methods (Section 1 of page 1).
- Clarifying the limitations of our approach (Section 5 of page 10).

We greatly appreciate the recognition of our work.  In particular, we thank the reviewers for acknowledging the **novelty of Laplace block and BCs encoding** (cFxe, m5AJ, zATM) , the **impressive results** (QKXd, m5AJ), the **thorough experiments** (QKXd, zATM) and the **clear and well-written paper** (QKXd, LbRf, m5AJ, zATM).

Please do feel free to let us know if you have any further questions.

Thank you once again for your time and thoughtful reviews.

Best regards,

The Authors of the Paper

---

### Comment · Area_Chair_izr3 · 2024-11-25
**Reviewers' Response**

Dear Reviewers,

As the author-reviewer discussion period is approaching its end, I would strongly encourage you to read the authors' responses and acknowledge them, while also checking if your questions/concerns have been appropriately addressed.

This is a crucial step, as it ensures that both reviewers and authors are on the same page, and it also helps us to put your recommendation in perspective.

Thank you again for your time and expertise.

Best,

AC

---

### Author Response · Authors · 2024-12-03
**Special thanks to all reviewers**

Dear Reviewers,

We sincerely thank you for your time, effort, and thoughtful feedback throughout the review process.  The discussion has been highly productive and fruitful, and your constructive suggestions have been invaluable in helping us improve the quality of our paper.

Once again, we deeply appreciate your dedication and valuable insights.

Best regards,

The Authors

---

### Meta-Review · Area_Chair_izr3 · 2024-12-20

**Metareview:**

The paper introduces PhyMPGN, a GNN-based approach for emulating solutions to PDEs on coarse, unstructured meshes. PhyMPGN integrates a learnable Laplace-Beltrami operator and a specialized boundary condition padding strategy at both physical and latent-space level. This is then coupled with ODE solvers for emulating time-dependent PDEs in the small data regime.

Regarding originality, incorporating a Laplace block into a solver is not entirely new (see [1]), though it is novel for GNN-based methods. The paper posits that ML-based methods can be orders of magnitude faster than classical solvers. However, the examples presented (2D, fairly simple PDEs in rather simple geometries, without accuracy normalization) do not fully support this claim. For instance, the paper compares runtime with COMSOL (which is not a state-of-the-art solver) without specifying the particular discretization algorithm or the hardware (CPU vs. GPU) used for the computation. Therefore, the claim of speed remains unclear. Furthermore, as the results are limited to 2D with relatively simple PDEs, it is not evident how the padding strategy could be applied to impose boundary conditions in realistic 3D CAD geometries. In addition, the method doesn't seem to generalize very well with finer meshes, thus it imposes several restrictions for locally refined meshes around sharp corners (where the solution can be singular). As such, the impact of this methodology to advance numerical simulation remains limited with the current information in the paper.

Despite these limitations, the reviewers find the core ideas interesting, well motivated with a rather clear exposition. The numerical experiments consider well-established baselines, although there are more recent and potentially more competitive baselines that could have been included. Consequently, I recommend acceptance.

[1] Nguyen et al. PARCv2: Physics-aware Recurrent Convolutional Neural Networks for Spatiotemporal Dynamics Modeling. ICML 2024.

**Additional Comments On Reviewer Discussion:**

Some of the reviewers raised some issues about the novelty of the paper, which the authors responded to, however, the reviewers were not completely aware of the full literature. Also concerns were raised about the generalization to finer meshes, which the authors, brushed off as out-of-scope, to which I disagree, as locally refined meshes are routinely used in numerical simulation. Also, issues about generalization to 3D were raised, to which the authors only provided a fairly plain answer, without considering all the difficulties of handling 3D meshes with sharp corners, and elongated structures.

---

### Decision · Program_Chairs · 2025-01-22

Accept (Spotlight)